# Whole-brain connectivity atlas of glutamatergic and GABAergic neurons in the mouse dorsal and median raphe nuclei

Zhengchao Xu[1], Zhao Feng[1,2], Mengting Zhao[1], Qingtao Sun[2], Lei Deng[1], Xueyan Jia[2], Tao Jiang[2], Pan Luo[1], Wu Chen[1], Ayizuohere Tudi[1], Jing Yuan[1,2], Xiangning Li[1,2], Hui Gong[1,2,3], Qingming Luo[1,2,4], Anan Li[1,2,3]*

[1]Britton Chance Center for Biomedical Photonics, Wuhan National Laboratory for Optoelectronics, MoE Key Laboratory for Biomedical Photonics, School of Engineering Sciences, Huazhong University of Science and Technology, Wuhan, China; [2]HUST-Suzhou Institute for Brainsmatics, JITRI, Suzhou, China; [3]CAS Center for Excellence in Brain Science and Intelligence Technology, Chinese Academy of Science, Shanghai, China; [4]School of Biomedical Engineering, Hainan University, Haikou, China

**Abstract** The dorsal raphe nucleus (DR) and median raphe nucleus (MR) contain populations of glutamatergic and GABAergic neurons that regulate diverse behavioral functions. However, their whole-brain input-output circuits remain incompletely elucidated. We used viral tracing combined with fluorescence micro-optical sectioning tomography to generate a comprehensive whole-brain atlas of inputs and outputs of glutamatergic and GABAergic neurons in the DR and MR. We found that these neurons received inputs from similar upstream brain regions. The glutamatergic and GABAergic neurons in the same raphe nucleus had divergent projection patterns with differences in critical brain regions. Specifically, MR glutamatergic neurons projected to the lateral habenula through multiple pathways. Correlation and cluster analysis revealed that glutamatergic and GABAergic neurons in the same raphe nucleus received heterogeneous inputs and sent different collateral projections. This connectivity atlas further elucidates the anatomical architecture of the raphe nuclei, which could facilitate better understanding of their behavioral functions.

*For correspondence:
aali@hust.edu.cn

Competing interest: The authors declare that no competing interests exist.

## Editor's evaluation

Using viral labeling method in combination with the fMOST imaging technology, the authors constructed a whole brain connectivity atlas of two subclasses, glutamatergic and GABAergic, of neurons in the dorsal raphe and median raphe nuclei. This study will be of interest to many neuroscientists who study neural circuits and cell type-specific functions.

## Introduction

The dorsal raphe nucleus (DR) and median raphe nucleus (MR, equivalent to the superior central nucleus raphe in the Allen Mouse Brain Common Coordinate Framework version 3 (Allen CCFv3)) belong to the rostral group of the raphe nuclei and contain most of brain's serotonergic neurons (*Wang et al., 2020*; *Watson et al., 2012*). The DR and MR are involved in a multitude of functions (*Domonkos et al., 2016*; *Huang et al., 2019*; *Szőnyi et al., 2019*); moreover, they have different,

and even antagonistic roles in the regulation of specific functions, including emotional behavior, social behavior, and aggression (*Balázsfi et al., 2018*; *Ohmura et al., 2020*; *Teissier et al., 2015*). The diverse regulatory processes are related to the connectivity of heterogeneous raphe groups (*Muzerelle et al., 2016*; *Nectow et al., 2017*; *Schneeberger et al., 2019*). Deciphering precise input and output organization of different neuron types in the DR and MR is fundamental for understanding their specific functions.

The DR and MR contain diverse neuron types, including glutamatergic, GABAergic, and serotonergic neurons, where the glutamatergic neurons mainly comprise of vesicular glutamate transporter 2 positive (Vglut2+) and 3 positive (Vglut3+) neurons (*Huang et al., 2019*; *Cardozo Pinto et al., 2019*; *Sos et al., 2017*). Several studies have revealed that the glutamatergic and GABAergic neurons in the DR and MR are involved in specific functions. In the DR, glutamatergic neurons play an important role in reward processing (*McDevitt et al., 2014*; *Liu et al., 2014*), while GABAergic neurons are involved in regulating energy expenditure (*Schneeberger et al., 2019*); moreover, they have opposite effects on feeding (*Nectow et al., 2017*). In the MR, glutamatergic neurons are critical for processing negative experiences, and activation of them induces aversive behavior, aggression and depressive symptoms (*Szőnyi et al., 2019*). Furthermore, MR GABAergic neurons are involved in regulating hippocampal theta rhythm, which is crucial for learning and memory (*Aitken et al., 2018*; *Li et al., 2005*). The diverse functions of specific neuron types in the raphe nuclei are highly dependent on their unique input-output circuits (*Ren et al., 2018a*). To have a more comprehensive understanding of the specific functions of glutamatergic and GABAergic neurons in the raphe nuclei, there is a need to determine how the cellular heterogeneity maps to whole-brain connectivity. Given that numerous Vglut3+ neurons in the DR and MR are also serotonergic (*Huang et al., 2019*; *Cardozo Pinto et al., 2019*; *Sos et al., 2017*), while Vglut2+ neurons in the DR and MR were distinct from the serotonergic neurons (*Huang et al., 2019*; *Szőnyi et al., 2019*), the present study focused on the connectivity of Vglut2+ neurons in the DR and MR.

Previous studies have revealed that the DR and MR integrate massive inputs from and send outputs to vast brain regions in the forebrain and midbrain (*Marcinkiewicz et al., 1989*; *Oh et al., 2014*; *Peyron et al., 1997*; *Vertes and Linley, 2008*). But these studies were unable to elucidate the neural connections of specific neuron types. Studies using slice physiological recording combined with optogenetics have demonstrated that DR GABAergic neurons receive long-range functional inputs from six upstream brain areas, including the prefrontal cortex, amygdala, lateral habenula (LH), lateral hypothalamic area (LHA), preoptic area, and substantia nigra (*Zhou et al., 2017*). However, optogenetic technology and physiological recording usually focus on specific regions connected with targeted neurons, making it difficult to dissect whole-brain long-range connections. Genetic targeting of neuronal subpopulations with Cre driver mouse and virus tracing make it possible to label the whole-brain long-range connectivity of specific neuron types (*Callaway and Luo, 2015*; *Huang and Zeng, 2013*; *Wickersham et al., 2007*). Several studies have revealed a portion of the long-range connections of glutamatergic and GABAergic neurons in the DR and MR through viral tracing techniques. For example, DR GABAergic neurons receive vast inputs and send projections to the dorsomedial nucleus of the hypothalamus (DMH) and bed nuclei of the stria terminalis (BST) (*Schneeberger et al., 2019*; *Weissbourd et al., 2014*). Moreover, MR Vglut2+ neurons are innervated by certain aversion/fear or memory-related areas, such as the LH, and send projections to the LH, medial ventral tegmental area, medial septum, and the vertical limbs of the diagonal bands of Broca (*Szőnyi et al., 2019*). Nevertheless, there is still a lack of whole-brain quantitative results and comprehensive analysis of the input-output circuits of glutamatergic and GABAergic neurons in the DR and MR. Furthermore, precise characterization and systematic quantitative analysis of whole-brain inputs and outputs require whole-brain high-resolution imaging of labeled neural structures and effective data processing methods to identify and integrate neural circuits.

In this study, we implemented a pipeline composed of viral tracing, whole-brain high-resolution imaging, data processing and analysis to dissect whole-brain inputs and outputs of glutamatergic and GABAergic neurons in the DR and MR and understand their organizational principle. We used modified monosynaptic rabies viral tracers to label the input neurons and enhanced yellow fluorescent protein (EYFP)-expressing adeno-associated virus (AAV) to trace whole-brain axon projections. Combined with home-made fluorescence micro-optical sectioning tomography (fMOST) (*Gong et al., 2016*), we acquired whole-brain datasets of labeled inputs and outputs at single-neuron resolution.

We identified the long-range input/output circuits, quantified the whole-brain distribution, analyzed the whole-brain connectivity pattern, and generated a precise whole-brain atlas of inputs and outputs of glutamatergic and GABAergic neurons in the DR and MR, which could facilitate the understanding of their functional differences and provide anatomical foundations for investigating into their functions. Furthermore, we developed the interactive website (http://atlas.brainsmatics.org/a/xu2011) to better present and share the raw data and results.

## Results

### Whole-brain mapping of monosynaptic input neurons to glutamatergic and GABAergic neurons in the DR and MR

To target glutamatergic and GABAergic neurons in the DR and MR, we used *Vglut2-Cre* (also known as *Slc17a6-Cre*) and *glutamate decarboxylase 2 (Gad2)-Cre* driver line mice. To verify the distribution pattern of Vglut2+ and Gad2+ neurons, we crossed the Cre driver line mice with reporter line mice (*Vglut2-Cre: LSL-H2B-GFP* mice and *Gad2-Cre: LSL-H2B-GFP* mice) (*Figure 1—figure supplement 1*). In the DR, Vglut2+ neurons were mostly found in the rostral part of the DR, while Gad2+ neurons were widely distributed and densely assembled in the lateral DR. In the MR, Vglut2+ neurons were mainly found in the caudal part of the MR, and the Vglut2+ neurons in the rostral part of the MR were mainly distributed laterally; moreover, Gad2+ neurons were distributed throughout the MR.

To label whole-brain inputs to glutamatergic and GABAergic neurons in the DR and MR, we used monosynaptic rabies tracing technique combined with *Vglut2-Cre* and *Gad2-Cre* driver line mice. First, Cre-dependent helper viruses, rAAV2/9-EF1α-DIO-His-TVA-BFP and rAAV2/9- EF1α-DIO-RG, were injected into the DR or MR. After 3 weeks, RV-ΔG-EnvA-GFP was injected into the same site (*Figure 1A*). The Cre-positive neurons infected by the Cre-dependent helper viruses could express the TVA receptor and glycoprotein. The rabies virus pseudotyped with the avian sarcoma leucosis virus glycoprotein EnvA could infect these neurons by binding TVA receptor specifically. Then, the rabies virus could be replenished with glycoprotein to retrogradely traverse to monosynaptic input neurons. The neurons co-labeled by blue and green fluorescent protein (BFP and GFP, respectively) in the injection sites were starter cells, and GFP-labeled neurons were input neurons (*Figure 1B and C*; *Figure 1—figure supplement 2A,B*). Most of the starter cells were within the injection site, with a fraction of the starter cells spreading to the neighboring areas (*Figure 1—figure supplement 2C*).

We performed in situ hybridization to characterize the specificity of labeled starter cells in the *Vglut2-Cre* mice and found that they were Vglut2 positive, with a few simultaneously being Vglut3 positive (*Figure 1B and C*; *Figure 1—figure supplement 3*), which was confirmed by immunohistochemical staining (*Figure 1—figure supplement 4*). To evaluate potential leakage expression of the virus, we performed control experiments in wild-type mice. As a result, there were few neurons infected by the AAV helper virus (BFP) and the RV (GFP) only at the injection site, but there were no GFP-labeled neurons in known upstream brain regions (*Figure 1—figure supplement 5*).

To acquire the whole-brain high-resolution datasets, the virus-labeled samples were embedded in glycol methacrylate (GMA) resin and imaged with our home-made fMOST system (*Gong et al., 2016*) at a resolution of 0.32 × 0.32 × 2 μm³ (*Figure 1D and E*). Such high-resolution images indicate that the soma and neurites of labeled input neurons are finely detailed. From anterior to posterior, we observed dense input neurons in the isocortex, striatum, pallidum, thalamus, hypothalamus, midbrain, pons, medulla, and cerebellar nuclei (*Figure 1—figure supplement 6*). Contrastingly, in the olfactory areas, cortical subplate, hippocampus, and cerebellar cortex, there were none or sparse input neurons (*Figure 1—figure supplement 6*).

### Quantified whole-brain inputs to glutamatergic and GABAergic neurons in the DR and MR

To quantify the distribution of monosynaptic input neurons in each brain region, we detected the coordinates of the soma of input neurons using the NeuroGPS algorithm (*Quan et al., 2013*) and manually checked them. The soma of input neurons were registered to the Allen CCFv3 (*Figure 2A,B*; Materials and methods) (*Ni et al., 2020*; *Wang et al., 2020*). Based on Allen CCFv3's hierarchy of brain regions, we identified 71 brain regions that have close connections with glutamatergic and GABAergic neurons in the DR and MR for subsequent analysis (Materials and methods; *Supplementary file 1*).

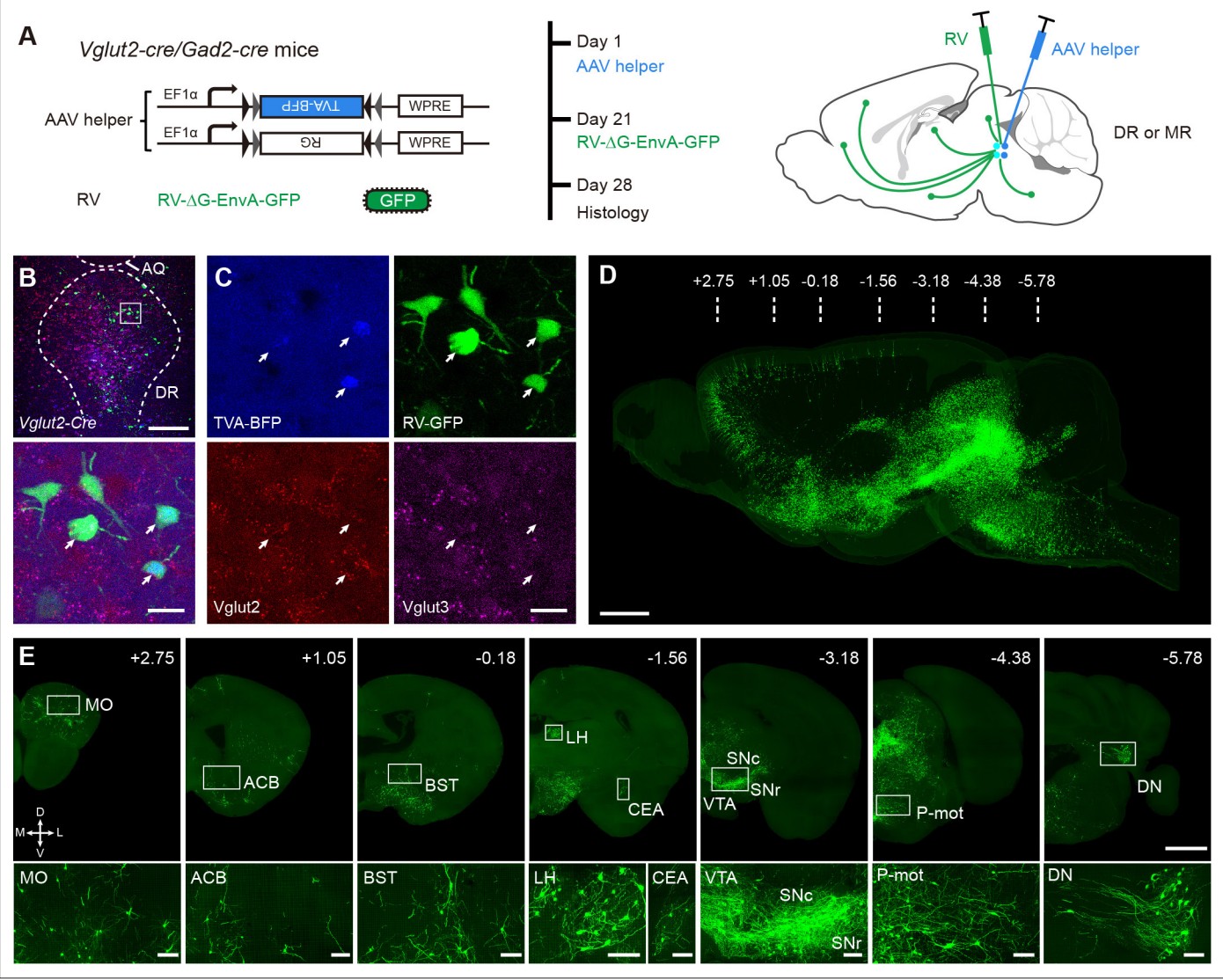

**Figure 1.** Whole-brain mapping of monosynaptic input neurons to cell-type-specific neurons in the DR and MR. (**A**) Schematic of monosynaptic rabies virus tracing the inputs to cell-type-specific neurons. The AAV helper virus expresses a fusion of TVA- BFP and RG, and the modified rabies virus pseudotyped with EnvA expresses GFP. The experimental strategy and time line are shown on the right. (**B**) Characterization of the specificity of starter cells at the DR in *Vglut2-Cre* mice using in situ hybridization. Bottom, enlarged view of the box area in the top image. Scale bar, top, 200 μm, bottom, 20 μm. (**C**) Detailed view of the bottom image in (**B**). White arrows, starter cells. (**D**) Three-dimensional rendering of whole-brain input neurons to DR glutamatergic neurons from a representative sample. Scale bar, 1 mm. (**E**) Representative coronal sections of maximum intensity projection showing the distribution of input neurons to DR glutamatergic neurons. The projections are 50 μm thick. Scale bars, top row, 1 mm, bottom row, 100 μm. A, anterior; P, posterior; M, medial; L, lateral. The details of abbreviations for brain regions see ***Supplementary file 1***.

The online version of this article includes the following figure supplement(s) for figure 1:

**Figure supplement 1.** Distribution and total number of Vglut2+ and Gad2+ neurons in the DR and MR.

**Figure supplement 2.** Validation of the labeling of whole-brain inputs.

**Figure supplement 3.** Characterization of the specificity of starter cells using in situ hybridization.

**Figure supplement 4.** Validation of the specificity of starter cells using immunohistochemical staining.

**Figure supplement 5.** Control experiments for mapping monosynaptic inputs.

**Figure supplement 6.** Representative images showing whole-brain inputs to the glutamatergic and GABAergic neurons in the DR and MR.

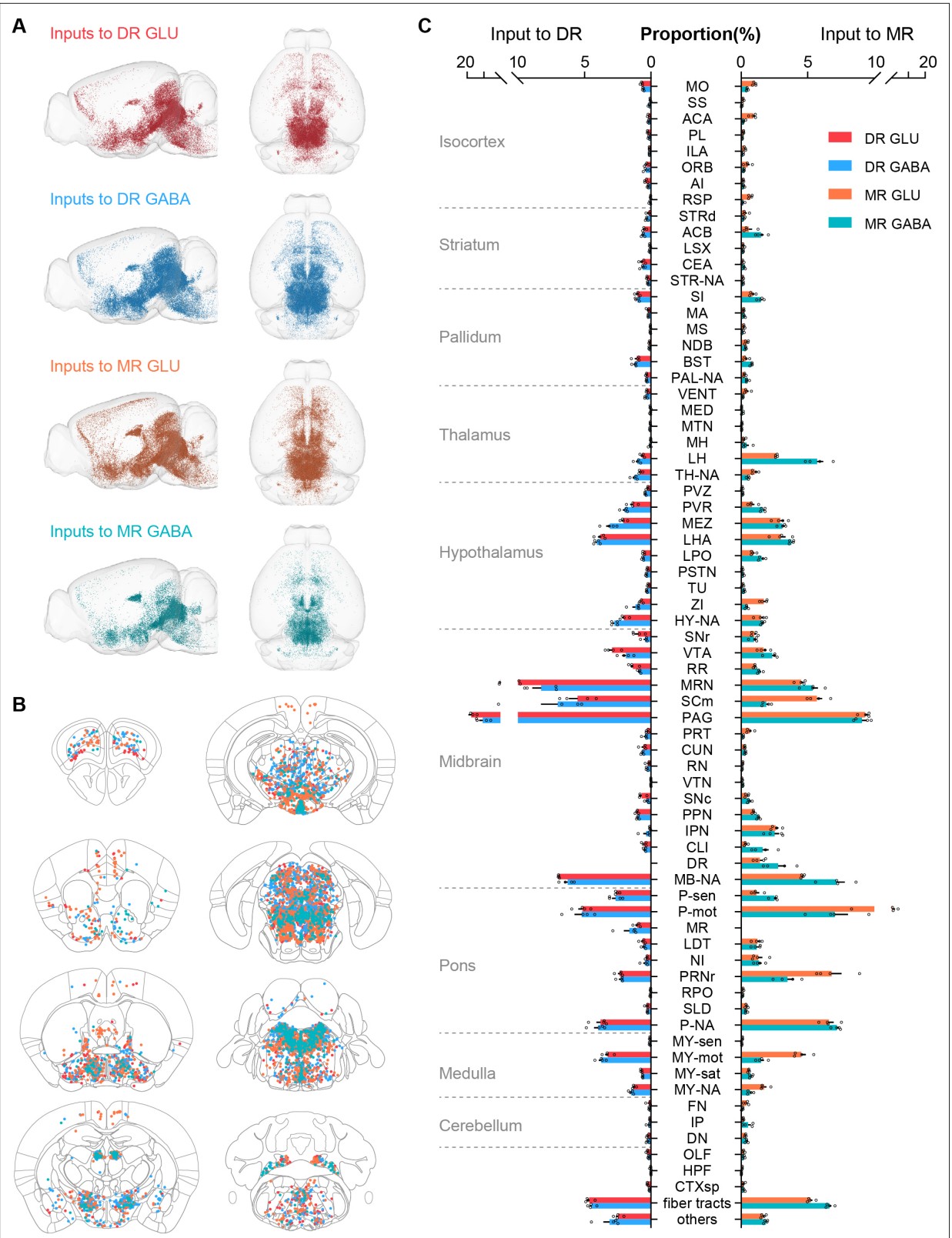

**Figure 2.** Whole-brain distribution of input neurons to glutamatergic and GABAergic neurons in the DR and MR. (**A**) Three-dimensional visualization of whole-brain inputs to glutamatergic neurons (GLU) and GABAergic neurons (GABA) in the DR and MR in representative samples. (**B**) Representative coronal sections illustrating the detected and registered input neurons. One dot represents one input neuron while different colors reflect inputs to different neuron types as in (**A**). Each section is 50 μm thick. (**C**) Proportion of the input neurons to glutamatergic and GABAergic neurons in the DR

*Figure 2 continued on next page*

*Figure 2 continued*

and MR across individual brain regions. Data are shown as mean ± s.e.m., n = 4 per group. The source data see *Supplementary file 2*. The details of abbreviations for brain regions see *Supplementary file 1*. The abbreviation NA indicates the non-annotated area in Allen CCFv3.

The online version of this article includes the following figure supplement(s) for figure 2:

**Figure supplement 1.** Visualization and comparison of whole-brain inputs across samples.

To generate the distribution of whole-brain input neurons, we calculated the number of input neurons in each brain region. To eliminate the variability in the total number of input neurons of different samples, the data were normalized by the total number of input neurons (with the exclusion of neurons in the injection site) to determine the proportion of input neurons in each brain region. Therefore, we quantified the whole-brain distribution of the long-range input neurons (*Figure 2C*; *Supplementary file 2*). To evaluate the across-sample consistency of the inputs to the same neuron group, we performed correlation analysis. The highly correlated results indicated the consistency and reliability of our data (*Figure 2—figure supplement 1A,B*). Moreover, we conducted unsupervised hierarchical clustering and bootstrapping of all samples. The input patterns of the four neuron groups were divided into two clusters based on the injection site, then the input patterns of MR glutamatergic and GABAergic neurons were segregated based on the neuron types (*Figure 2—figure supplement 1C*).

## Comparison of inputs to glutamatergic and GABAergic neurons in the DR and MR

To explore the relationship of whole-brain long-range inputs to glutamatergic and GABAergic neurons in the DR and MR, we initially compared the inputs from the MR to glutamatergic and GABAergic neurons in the DR and found no significant difference (p = 0.222, one-way ANOVA); then we compared the inputs from the DR to glutamatergic and GABAergic neurons in the MR and also found no significant difference (p = 0.069, one-way ANOVA). Next, we compared the whole-brain inputs to glutamatergic and GABAergic neurons in the DR and MR across brain regions using correlation analysis and variance analysis (one-way ANOVA followed by multiple comparisons with Tukey's test; *Supplementary file 2*; *Ogawa et al., 2014*). There were highly similar whole-brain inputs to glutamatergic and GABAergic neurons in the same raphe nucleus. Contrastingly, there were similar whole-brain inputs to the same neuron type in the DR and MR, with relatively lower correlation coefficients (*Figure 3A–D*). Furthermore, there were quantitative differences in certain brain regions embedded in the overall similarity of the input patterns (*Figure 3A–D*).

Specifically, a modest proportion of input neurons were distributed in the isocortex (*Figure 2C*). Moreover, several brain regions had biased inputs to different raphe neuron groups, especially the somatomotor areas (MO), anterior cingulate area (ACA) and retrosplenial area (RSP), which preferentially innervated MR glutamatergic neurons in comparison with MR GABAergic neurons and DR glutamatergic neurons (*Figure 3A–D*).

The striatum and pallidum had considerable inputs to glutamatergic and GABAergic neurons in the DR and MR (*Figure 2C*). Notably, the central amygdalar nucleus (CEA) preferentially innervated glutamatergic and GABAergic neurons in the DR than those in the MR (*Figure 3C–E*). And the BST sent prominent inputs to glutamatergic and GABAergic neurons in the DR and MR, with a preference for the DR (*Figure 3C and D*). Compared with the glutamatergic and GABAergic in the DR, those in the MR received a larger proportion of inputs from the diagonal band nucleus (NDB) (*Figure 3C and D*).

The majority of input neurons in the thalamus were located in the LH (*Figure 2C*). Additionally, there was a preference for LH neurons to have more inputs to MR glutamatergic and GABAergic neurons than to DR neurons (*Figure 3C and D*). Notably, MR GABAergic neurons received more inputs from the LH than MR glutamatergic neurons (*Figure 3B*). Moreover, there were vast inputs from the hypothalamus to DR and MR glutamatergic and GABAergic neurons, with the LHA providing the largest proportion, followed by the hypothalamic medial zone (MEZ), periventricular region (PVR), zona incerta (ZI), and lateral preoptic area (LPO) (*Figure 2C*). The ZI provided more inputs to MR glutamatergic neurons than to MR GABAergic neurons and DR glutamatergic neurons (*Figure 3B and C*). The LPO preferentially innervated MR GABAergic neurons in comparison with DR GABAergic neurons and MR glutamatergic neurons (*Figure 3B and D*).

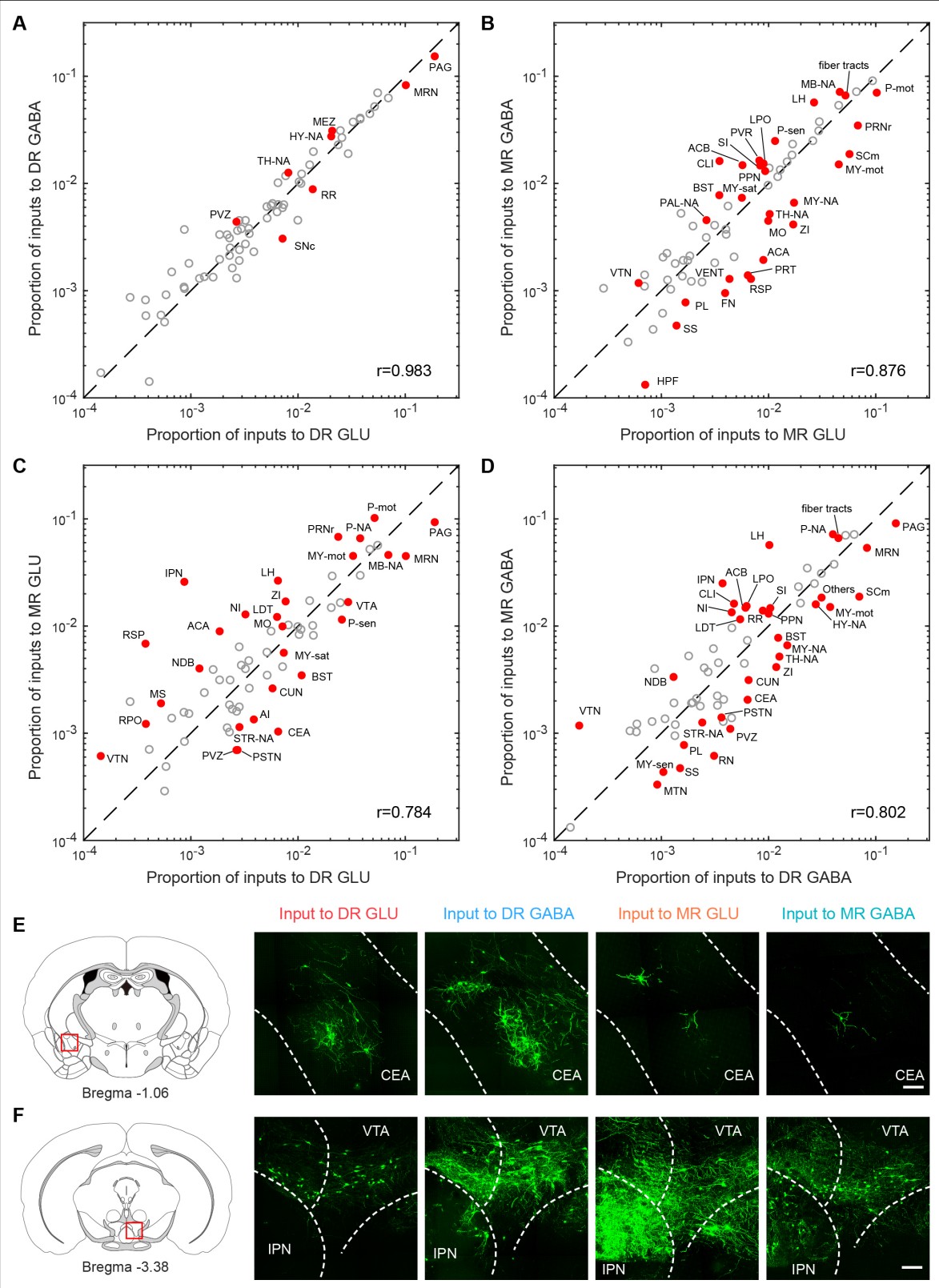

**Figure 3.** Comparisons of inputs to glutamatergic and GABAergic neurons in the DR and MR. (**A**) Comparison between inputs to the glutamatergic and GABAergic neurons in the DR. (**B**) Comparison between inputs to the glutamatergic and GABAergic neurons in the MR. (**C**) Comparison between inputs to the glutamatergic neurons in the DR and MR. (**D**) Comparison between inputs to the GABAergic neurons in the DR and MR. The circles represent the proportion of input neurons in each brain region, where red and solid circles indicate significant differences ($p < 0.05$, One-way ANOVA followed

*Figure 3 continued on next page*

*Figure 3 continued*

by multiple comparisons with Tukey's test). The p-values see *Supplementary file 2*. r: Pearson's correlation coefficients. The details of abbreviations for brain regions see *Supplementary file 1*. (**E**) Comparison of input neurons in the CEA. Left: position of the images on the right. Right: RV-GFP-labeled input neurons in the CEA. Representative images are from maximum intensity projections of the coronal sections. The projections were 50 μm thick. Scale bar, 100 μm. (**F**) Comparison of input neurons in the IPN. Left: position of the images on the right. Right: RV-GFP-labeled input neurons in the IPN. Representative images are from maximum intensity projections of the coronal sections. The projections were 50 μm thick. Scale bar, 100 μm.

The midbrain had the largest proportion of inputs to glutamatergic and GABAergic neurons in the DR and MR (*Figure 2C*). Although glutamatergic and GABAergic neurons in the DR and MR received massive inputs from the periaqueductal gray (PAG) and midbrain reticular nucleus (MRN), DR neurons received more than MR neurons (*Figure 3C and D*). The interpeduncular nucleus (IPN) provided remarkable inputs to glutamatergic and GABAergic neurons in the MR with very sparse inputs to the DR (*Figure 3C, D and F*). The superior colliculus, motor related (SCm) contributed more inputs to MR glutamatergic neurons and DR GABAergic neurons than to MR GABAergic neurons (*Figure 3B and D*).

The pons contributed dense inputs. Furthermore, the pons, motor related (P-mot) and pontine reticular nucleus (PRNr) preferentially innervated MR glutamatergic neurons in comparison with MR GABAergic neurons and DR glutamatergic neurons (*Figure 3B and C*). However, the pons, sensory related (P-sen) provided more inputs to DR glutamatergic neurons than to MR glutamatergic neurons (*Figure 3C*). Moreover, the medulla, motor related (MY-mot) preferentially provided inputs to MR glutamatergic neurons in comparison with MR GABAergic neurons and DR glutamatergic neurons (*Figure 3B and C*). These findings indicated that the glutamatergic and GABAergic neurons in the DR and MR received inputs from similar upstream brain regions with quantitative differences in specific brain regions.

## Whole-brain outputs of glutamatergic and GABAergic neurons in the DR and MR

To systematically map whole-brain outputs of glutamatergic and GABAergic neurons in the DR and MR, we stereotaxically injected Cre-dependent AAV-DIO-EYFP into the DR or MR in *Vglut2-Cre* and *Gad2-Cre* mice (n = 4 per group). The virus-labeled and GMA resin-embedded samples were imaged using fMOST system (*Figure 4A*). To generate whole-brain quantified outputs, we registered the high-resolution whole-brain image datasets to Allen CCFv3, and segmented the injection site and projection signal to calculate the proportion of projection signal across brain regions (*Figure 4A and B*; *Figure 4—figure supplement 1*; *Supplementary file 3*; Materials and methods).

At the whole-brain level, glutamatergic and GABAergic neurons in the DR and MR had substantial ascending projections to the forebrain and midbrain and varying degrees of descending projections to the pons and medulla (*Figure 4B*; *Figure 4—figure supplement 2*). MR glutamatergic and GABAergic neurons predominately innervated midline structures, while DR glutamatergic and GABAergic neurons sent more broad and lateral projections, and their projection targets were largely distinctive. Regarding the forebrain, DR neurons projected more broadly to the prefrontal cortex, amygdala, nucleus accumbens (ACB) and BST, while MR neurons innervated the lateral septal complex, medial septal nucleus (MS), NDB and LH (*Figure 4B*; *Figure 4—figure supplement 2*). And glutamatergic and GABAergic neurons in the MR had more outputs to the pons than those in the DR (*Figure 4B*). Meanwhile, they both sent dense projections to the hypothalamus and midbrain areas, such as the LHA and ventral tegmental area (VTA) (*Figure 4B*; *Figure 4—figure supplement 2*). Moreover, glutamatergic neurons project more broadly than GABAergic neurons. In contrast, GABAergic neurons preferentially innervated neighboring brain regions, such as the PAG and MRN for DR GABAergic neurons and the IPN for MR GABAergic neurons.

We quantitatively compared the projection patterns of glutamatergic and GABAergic neurons in the DR and MR. The same neuron types in the DR and MR had divergent projection patterns (*Figure 4C*). Regarding the glutamatergic and GABAergic neurons in the same raphe nucleus, although their overall projection patterns were relatively similar, there were differences in critical brain regions (*Figure 4C–E*; *Supplementary file 3*). Notably, regarding the amygdala, DR GABAergic neurons mainly projected to the CEA, with scarce projections to the basolateral amygdalar nucleus (BLA); contrastingly, DR glutamatergic neurons preferentially projected to the BLA (*Figure 4—figure supplement 3A,D*). And

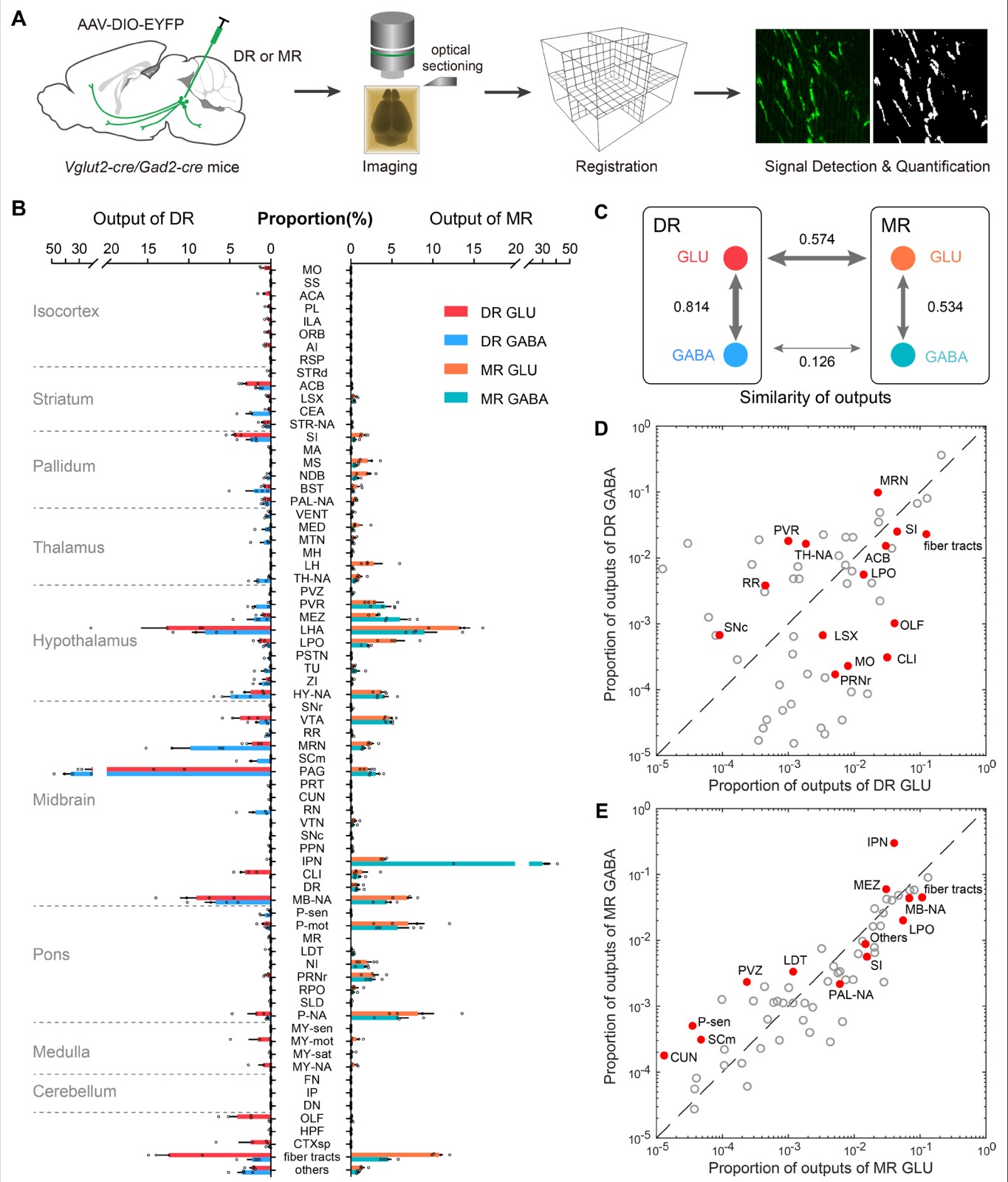

**Figure 4.** Whole-brain outputs of glutamatergic and GABAergic neurons in the DR and MR. (**A**) Schematic outlining viral tracing, whole-brain imaging, data processing and analysis. (**B**) Proportion of the outputs of glutamatergic and GABAergic neurons in the DR and MR across individual brain regions. Data are shown as mean ± s.e.m., n = 4 per group. The source data see *Supplementary file 3*. (**C**) Similarities of whole-brain output patterns. The numbers indicate Pearson's correlation coefficients. The arrow thickness indicates the magnitude of similarity. (**D**) Comparison between outputs of

*Figure 4 continued on next page*

*Figure 4 continued*

glutamatergic and GABAergic neurons in the DR. (**E**) Comparison between outputs of glutamatergic and GABAergic neurons in the MR. The circles represent the proportion of outputs in each target brain region, where red and solid circles indicate significant differences (p < 0.05, one-way ANOVA). The p-values see *Supplementary file 3*. The details of abbreviations for brain regions see *Supplementary file 1*.

The online version of this article includes the following figure supplement(s) for figure 4:

**Figure supplement 1.** Validation of the injection sites of whole-brain outputs.

**Figure supplement 2.** Representative images showing projections from glutamatergic and GABAergic neurons in the DR and MR.

**Figure supplement 3.** Comparison of outputs from glutamatergic and GABAergic neurons in the DR.

DR GABAergic neurons sent considerable projections to the DMH and paraventricular nucleus of the thalamus (PVT), while there were scarce or no axonal projections of DR glutamatergic neurons in these regions (*Figure 4—figure supplement 3B-D*). Regarding the MR, there were dense projections of glutamatergic neurons in the LH but scarce projections of GABAergic neurons (*Figure 4B*). Furthermore, the IPN received 29.9 % of the total projections from MR GABAergic neurons with only 4.0 % of the total projections from MR glutamatergic neurons (*Figure 4B and E*).

## Habenula-Raphe circuits

The habenula, which comprises of the medial habenula (MH) and lateral habenula (LH), appears to be a node connecting the forebrain and midbrain regions that are related to emotional behaviors (*Hikosaka, 2010*). The LH has been closely connected to the DR and MR both anatomically and functionally. And their connections are involved in aversion-related behavior and depression (*Hu et al., 2020*; *Zhao et al., 2015*). The LH provided dense inputs to glutamatergic and GABAergic neurons in the DR and MR, with a preference for MR neurons than corresponding DR neurons (*Figure 3C and D*; *Figure 5A*). And the input neurons to MR glutamatergic and GABAergic neurons were assembled more caudally (*Figure 5A*). Specifically, MR GABAergic neurons received more inputs from the LH than MR glutamatergic neurons (*Figure 3D*), and we found that the lateral part of LH sent dense inputs to MR GABAergic neurons but sparser inputs to MR glutamatergic neurons (*Figure 5—figure supplement 1A,B*). Regarding the LH, the input neurons to MR GABAergic neurons seemed to be distributed more laterally than the input neurons to MR glutamatergic neurons on the whole (*Figure 5—figure supplement 1C*).

However, there were no projections from DR glutamatergic and GABAergic neurons and scarce projections from MR GABAergic neurons to the LH. Specifically, only MR glutamatergic neurons sent strong projections to the LH (mainly assembled in the medial part of LH) (*Figure 4B*; *Figure 5—figure supplement 1D-F*). Taking advantage of our three-dimensional high-resolution imaging, we found that MR Vglut2+ neurons sent projections to the LH through the fasciculus retroflexus, stria medullaris, and thalamus respectively (*Figure 5B*; *Figure 5—figure supplement 1E,F*). There was evidence that Vglut2+ neurons in surrounding regions of the MR did not project to the LH (*Szőnyi et al., 2019*), which confirmed the reliability of this projection pattern. The specific reciprocal connections between MR glutamatergic neurons and the LH suggested that MR glutamatergic neurons might be involved in specific functions related to the LH. The LH has been revealed to play a critical role in aversion and depression (*Cui et al., 2018*; *Hu et al., 2020*; *Yang et al., 2018*). Moreover, MR Vglut2+ neurons could activate the LH, and activation of MR Vglut2+ neurons could induce aversive behaviors and depressive symptoms (*Szőnyi et al., 2019*). These results highlight the importance of the structural characteristics of the MR-LH pathway for their function roles.

Although the MH sent few inputs to MR glutamatergic and GABAergic neurons and scarce inputs to DR glutamatergic and GABAergic neurons, it is considered to strongly project to the IPN (*Lima et al., 2017*; *Qin and Luo, 2009*). In our results, the IPN had remarkable inputs to MR glutamatergic and GABAergic neurons but very sparse inputs to DR glutamatergic and GABAergic neurons (*Figure 3F*). These suggest that MR glutamatergic and GABAergic neurons receive inputs directly and indirectly (via the IPN) from the MH. Moreover, MR glutamatergic and GABAergic neurons strongly projected to the IPN (*Figure 5B*), which has been revealed to project to the LH (*Lima et al., 2017*). These results indicated the sophisticated connections of the habenula, IPN, DR, and MR. Based on the conventional model of the habenula-raphe circuit (*Hikosaka, 2010*; *Hu et al., 2020*), we proposed a more refined model of the habenula-raphe circuit (*Figure 5C*).

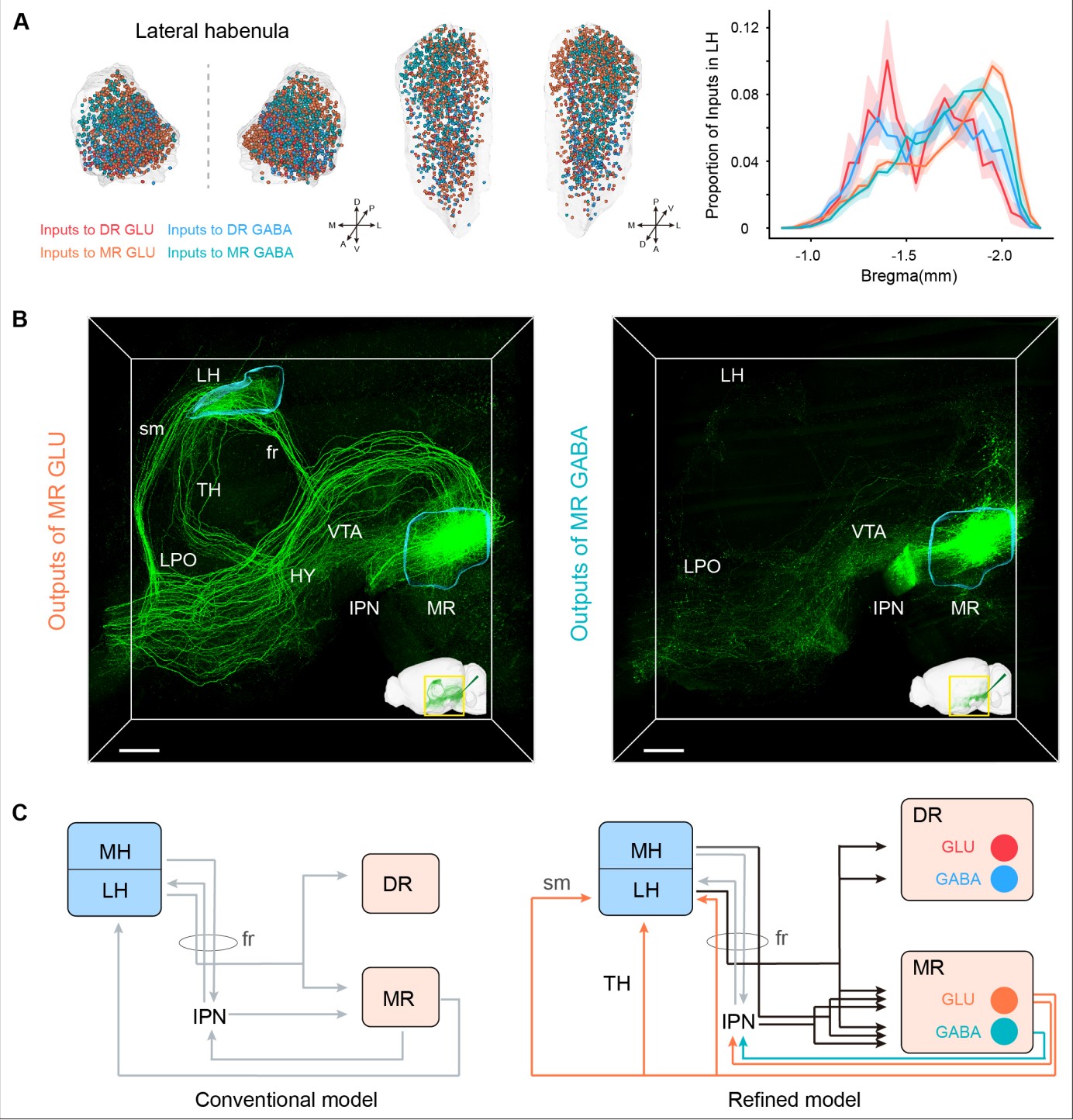

**Figure 5.** Habenula-raphe circuit. (**A**) Comparison of inputs in the LH to glutamatergic and GABAergic neurons in the DR and MR. Left and middle Coronal view and horizonal view of three-dimensional rendering of input neurons in the LH from representative samples. One dot represents one input neuron while different colors reflect inputs to different neuron types. Right, density plot of input neurons in the LH along the anterior-posterior axis. Bin width, 50 µm. The shaded area indicates s.e.m., n = 4. (**B**) Representative projections of MR glutamatergic and GABAergic neurons. The image is a perspective view of three-dimensional rending of projections in the region of interest shown in the bottom right corner. The image in the bottom right corner is three-dimensional rending of projections in the left hemisphere. The rendered data were registered to Allen CCFv3. Scale bar, 500 µm. (**C**) A refined model of the habenula-raphe circuit based on connections with glutamatergic and GABAergic neurons in the DR and MR. The conventional model is from previous studies (*Hikosaka, 2010*; *Hu et al., 2020*). In the refined model, the inputs identified in this study are shown in black, and the outputs of MR glutamatergic and GABAergic neurons are shown in orange and turquoise, respectively; moreover, the known circuits are shown in gray.

*Figure 5 continued on next page*

*Figure 5 continued*

TH, thalamus; HY, hypothalamus; sm, stria medullaris; fr, fasciculus retroflexus.

The online version of this article includes the following figure supplement(s) for figure 5:

**Figure supplement 1.** Representative inputs and outputs of glutamatergic and GABAergic neurons in the MR.

## Whole-brain connectivity pattern of glutamatergic and GABAergic neurons in the DR and MR

The glutamatergic and GABAergic neurons in the DR and MR received inputs from and sent outputs to a wide range of brain regions (*Figure 6—figure supplement 1A*). We assessed the similarities between the whole-brain inputs and outputs of the same neuron group using Pearson's correlation coefficient. Regarding glutamatergic and GABAergic neurons in the DR, the correlation coefficients were 0.766 and 0.839, respectively (*Figure 6—figure supplement 1A*). These results indicated that they had reciprocal connections with vast brain regions. Regarding glutamatergic and GABAergic neurons in the MR, the correlation coefficients were 0.578 and 0.384, respectively (*Figure 6—figure supplement 1A*), which was related to the fact that MR neurons received massive inputs from but sent sparse projections to the isocortex, striatum, and medulla (*Figure 2C*; *Figure 4B*).

There were massive reciprocal connections of glutamatergic and GABAergic neurons in the DR and MR, implicating feedback regulation of specific functions. To assess the reciprocity, we calculated the ratio of outputs to inputs for each brain region (*Figure 6—figure supplement 1B*). Approximately 45 % of brain regions had ratio values ranging from 0.25 to 4, which indicated relatively balanced reciprocal connectivity. Contrastingly, approximately 45 % of brain region had input bias (ratio value <0.25), with only a few brain regions showing output bias (ratio value >4). Particularly, for MR GABAergic neurons, the IPN accounted for 2.5 % of all inputs but 29.9 % of all outputs.

A vast range of upstream brain regions send inputs to the glutamatergic and GABAergic neurons in the DR and MR, whereas their relationships remain to be explored. And the axons of neurons have collateral branches targeting different areas, but the relationships between these targets remain unclear. Since the DR and MR were heterogeneous and each injection only labeled a portion of neurons that might have different connectivity, we performed correlation analysis and hierarchical cluster analysis to explore the similarities and variances of inputs and outputs of brain regions connected with glutamatergic and GABAergic neurons in the DR and MR. We selected 14 brain regions that have close long-range connections with glutamatergic and GABAergic neurons in the DR and MR for analysis. As a result, the clusters were not completely consistent regarding the inputs or outputs of glutamatergic and GABAergic neurons in either the DR or MR (*Figure 6A and B*). Regarding the inputs to DR glutamatergic and GABAergic neurons, upstream brain regions formed clear but inconsistent clusters, where only the substantia nigra, reticular part (SNr), substantia nigra, compact part (SNc), and VTA formed the same cluster. Regarding the outputs of DR glutamatergic neurons, the MEZ, LPO, LHA, ZI, SNr, and IPN formed a cluster with strong correlations. Contrastingly, different clusters were presented for outputs of DR GABAergic neurons. This suggested that DR glutamatergic and GABAergic neurons might have different collateral projection patterns. Additionally, a pair of brain regions might display opposing correlations in terms of the inputs or outputs of glutamatergic and GABAergic neurons in the DR. For example, the substantia innominate (SI) and NDB were positively correlated for inputs to DR glutamatergic neurons but negatively correlated for inputs to DR GABAergic neurons (*Figure 6A*), which suggested that the basal forebrain might send distinct inputs to heterogeneous DR subpopulations. There were clear clusters for the inputs and outputs of MR glutamatergic neurons. Specifically, regarding the clusters of inputs to MR glutamatergic neurons, the ZI was separate from other regions, indicating that the ZI-MR glutamatergic neurons pathway might execute special functions different from other upstream brain regions. There were no obvious clusters for the inputs and outputs of MR GABAergic neurons, suggesting that MR GABAergic neurons might have few collateral projections to these brain regions. Moreover, there were pairs of brain regions displaying opposing correlations regarding the inputs or outputs of MR glutamatergic and GABAergic neurons. For example, the SNc and IPN were positively correlated for inputs to MR glutamatergic neurons but negatively correlated for inputs to MR GABAergic neurons (*Figure 6B*). Overall, our findings indicated that glutamatergic and GABAergic neurons in the DR and MR might receive inputs from and project to various unions of brain regions.

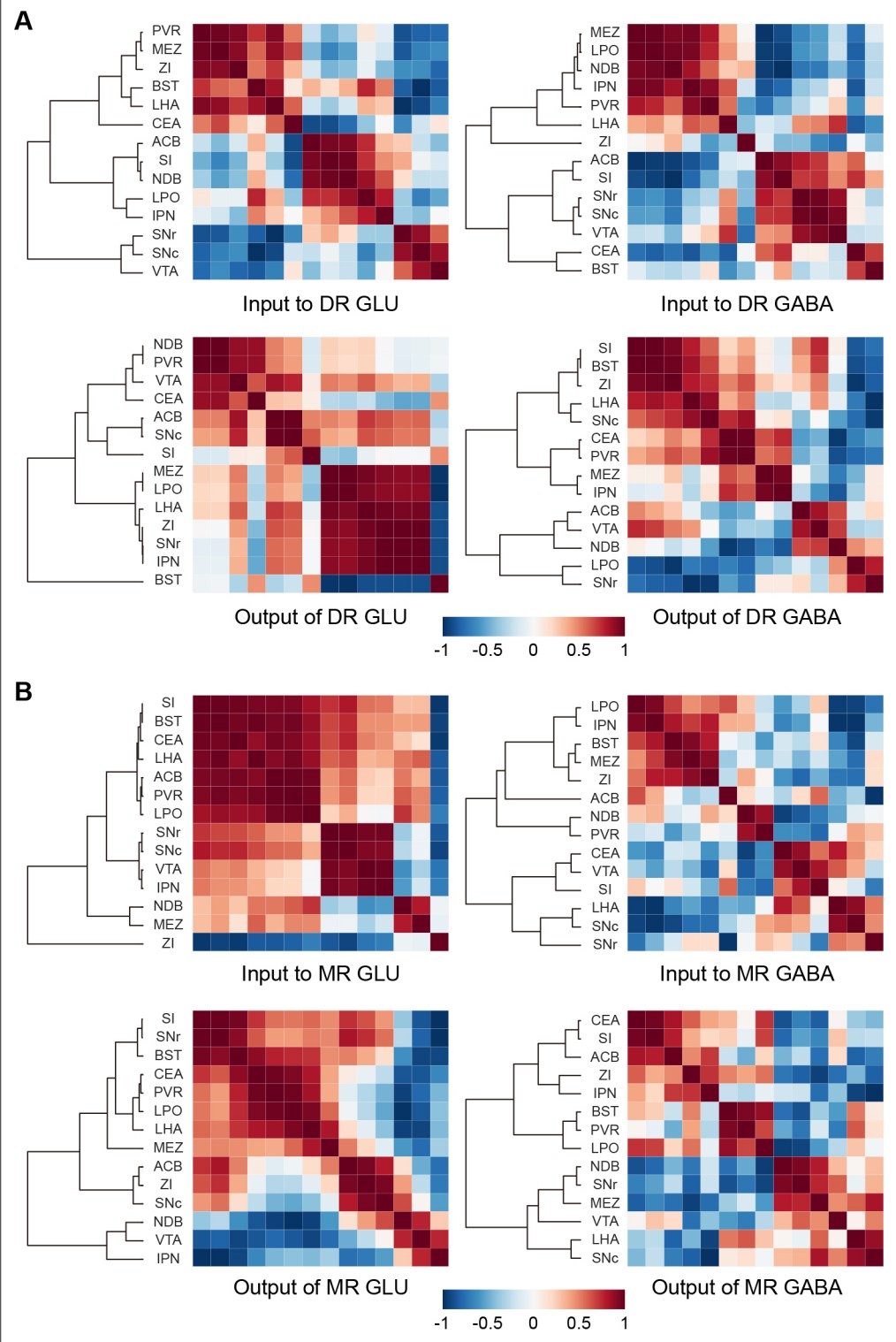

**Figure 6.** Connectivity patterns of glutamatergic and GABAergic neurons in the DR and MR. (**A**) Correlation and hierarchical cluster analysis showing the similarities and variances in brain regions connected with DR glutamatergic and GABAergic neurons. The heatmap represents Pearson's correlation coefficient matrix. (**B**) Correlation and hierarchical cluster analysis showing the similarities and variances in brain regions connected with MR glutamatergic and GABAergic neurons. The heatmap represents Pearson's correlation coefficient matrix. The details of abbreviations for brain regions see **Supplementary file 1**.

*Figure 6 continued on next page*

*Figure 6 continued*

The online version of this article includes the following figure supplement(s) for figure 6:

**Figure supplement 1.** Connectivity pattern of whole-brain inputs and outputs of glutamatergic and GABAergic neurons in DR and MR.

## Discussion

In this study, we used virus tracing and whole-brain high-resolution imaging to generate a comprehensive whole-brain atlas of inputs and outputs of glutamatergic and GABAergic neurons in the DR and MR (*Figure 7A and B*). Further, we performed systematic quantitative analysis to elucidate the convergence and divergence in the input and output patterns. Moreover, we proposed a more refined model of the habenula-raphe circuit based on the conventional model (*Hikosaka, 2010.*; *Hu et al., 2020*).

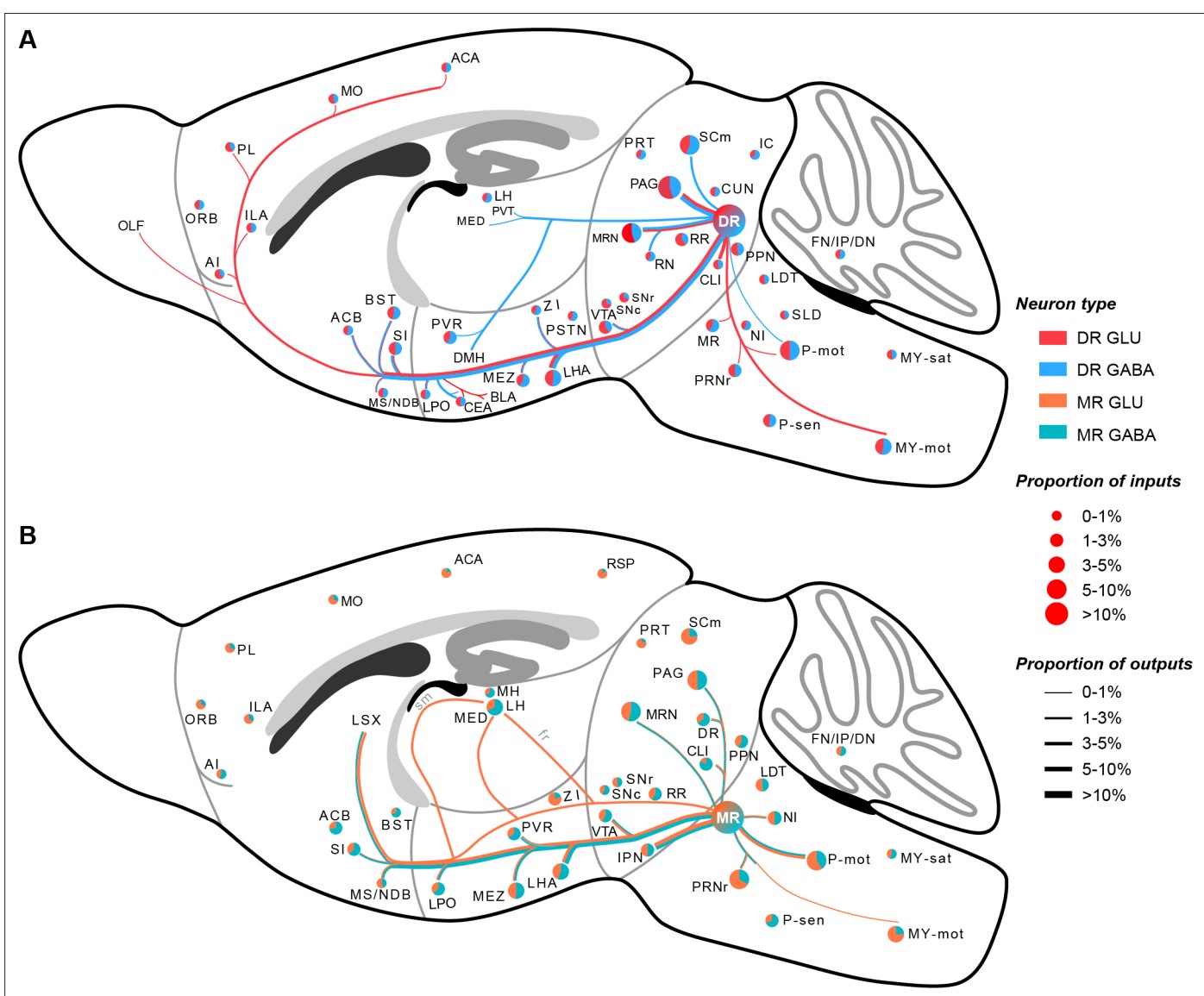

**Figure 7.** Whole-brain schematic of the inputs and outputs of glutamatergic and GABAergic neurons in the DR and MR. (**A**) Whole-brain schematic of the inputs and outputs of glutamatergic and GABAergic neurons in the DR. (**B**) Whole-brain schematic of the inputs and outputs of glutamatergic and GABAergic neurons in the MR. The pie charts represent the inputs in each brain region, where colors reflect the postsynaptic neuron types, and the size reflects the proportion value. The lines represent outputs in each brain region, where colors reflect the neuron types, and the line thickness reflects the proportion value. The details of abbreviations for brain regions see *Supplementary file 1*.

Our results are consistent with the input and output circuits of neurons in the DR and MR determined through classic tracing techniques (*Marcinkiewicz et al., 1989*; *Oh et al., 2014*; *Peyron et al., 1997*; *Vertes and Linley, 2008*). However, the DR and MR are heterogeneous and contain diverse neuron types, including Vglut2+, Vglut3+, GABAergic, and serotonergic neurons, where numerous Vglut3+ neurons are also serotonergic (*Huang et al., 2019*; *Cardozo Pinto et al., 2019*; *Sos et al., 2017*). Compared with the known circuits of specific neuron types in the DR and MR, our results are consistent with the current incomplete knowledge of the input and output circuits of Vglut2+ and GABAergic neurons in the DR and MR. Additionally, we found that different neuron types in the same raphe nucleus received inputs from similar upstream brain regions and sent complementary projections (*Lin et al., 2020*; *Muzerelle et al., 2016*; *Ogawa et al., 2014*; *Pollak Dorocic et al., 2014*; *Ren et al., 2018a*; *Ren et al., 2019*; *Szőnyi et al., 2019*; *Weissbourd et al., 2014*). Regarding the DR, compared with Vglut2+ and Gad2+ neurons, the serotonergic neurons had more broad projections, even extending to the entorhinal area and piriform area (*Ren et al., 2018a*; *Ren et al., 2019*). Notably, DR serotonergic neurons sent projections to the LH (*Muzerelle et al., 2016*; *Ren et al., 2018a*; *Zhang et al., 2018*), while DR Vglut2+ neurons and GABAergic neurons did not (*Figure 4B*). Regarding the MR, previous studies have revealed that neurons in the MR project to the hippocampus and regulate multiple hippocampal activities (*Jackson et al., 2008*; *Varga et al., 2009*; *Vertes and Linley, 2008*). However, we did not observe apparent projections of MR Vglut2+ and Gad2+ neurons to the hippocampus (*Figure 4B*), which was in consistent with the previous studies that most retrogradely labeled MR input neurons to the hippocampus were serotonergic or Vglut3-positive (*Szőnyi et al., 2016*). Besides, MR serotonergic neurons scarcely sent projections to the LH, the same as Vglut3+ neurons and GABAergic neurons (*Szőnyi et al., 2019*), but the Vglut2+ neurons sent dense projections to the LH through multiple pathways (*Figure 5B*). Furthermore, compared with the distribution of Vglut3+ and serotonergic neurons (*Lein et al., 2007*; *Oh et al., 2014*.), there were obvious differences among the overall distribution patterns of Vglut2+, Gad2+, and Vglut3+/serotonergic neurons in the same raphe nucleus (*Figure 1—figure supplement 1*), which might be relevant to the differences in connectivity. These biases resulting from cell-type specificity emphasize the necessity of dissecting the anatomical organization of different cell types in the same region.

Furthermore, there were differences between connections of glutamatergic and GABAergic neurons in the same raphe nucleus, which might provide insight into their functions. Unlike DR glutamatergic neurons, DR GABAergic neurons preferentially projected to the CEA (*Figure 4—figure supplement 3A,D*). Food intake is inhibited and increased by activation of the CEA and DR GABAergic neurons, respectively (*Carter et al., 2013*; *Nectow et al., 2017*.) Furthermore, DR GABAergic neurons uniquely innervate the PVT (*Figure 4—figure supplement 3C,D*), which is connected with the CEA and involved in inhibiting food intake (*Kirouac, 2015*). Therefore, we speculate that activation of DR GABAergic neurons might inhibit CEA and PVT neurons to increase food intake. DR glutamatergic and GABAergic neurons had scarce and considerable projections to the DMH, respectively. (*Figure 4—figure supplement 3B,D*). And DR GABAergic neurons have been revealed to regulate thermogenesis via projections to the DMH (*Schneeberger et al., 2019*). These results suggest that DR GABAergic neurons might play a more critical role in regulating thermogenesis than DR glutamatergic neurons. Moreover, we found that MR glutamatergic neurons projected to the LH through multiple pathways while MR GABAergic neurons scarcely projected to the LH (*Figure 5B*), which indicated that there might be different subtypes of MR glutamatergic neurons projecting to LH through different pathways. There is evidence that MR Vglut2+ neurons activate the LH (related to aversion and negative prediction) and control the acquisition of negative experience (*Szőnyi et al., 2019*). Our results suggest that MR glutamatergic neurons that project to the LH through multiple pathways might regulate the different aspects of aversive and negative emotions. Taken together, these results highlight that the biased or unique connectivity of different neuron types in the same raphe nucleus are related to the regulation of specific functions. Our findings could facilitate further elucidation of the functions of glutamatergic and GABAergic neurons in the DR and MR.

There were key differences between the connections of specific neuron types in the DR and MR. DR glutamatergic and GABAergic neurons had close connections with the CEA; contrastingly, MR glutamatergic and GABAergic neurons received few inputs from the CEA and did not project to the CEA (*Figure 6—figure supplement 1A*). And the CEA has been revealed to regulate reward and food intake (*Carter et al., 2013*; *Janak and Tye, 2015*; *Zséli et al., 2018*), which are also regulated

by DR glutamatergic and GABAergic neurons (*Nectow et al., 2017*). The IPN had dense reciprocal connections with MR glutamatergic and GABAergic neurons, but almost no direct connections with DR glutamatergic and GABAergic neurons (*Figure 6—figure supplement 1A*). The IPN and MR are important parts of the midline network involved in regulating the hippocampal theta rhythm (*Lima et al., 2017*). The consistency between anatomical connectivity and behavioral function indicates the significance of dissecting whole-brain connectivity for elucidating the functions.

Notably, a pair of brain regions might display different or even opposing correlations in terms of the inputs to glutamatergic and GABAergic neurons in the same raphe nucleus (*Figure 6A and B*), which suggests that different neuron groups in upstream brain regions might individually target heterogeneous raphe groups. There is a need for more advanced labeling methods that could label upstream inputs to different cell types in a single brain sample to further explore this problem. The projection target regions of glutamatergic and GABAergic neurons in the same raphe nucleus also showed different correlations (*Figure 6A and B*). This suggests that glutamatergic and GABAergic neurons in the same raphe nucleus might have different collateral projection patterns, which are worth illustrating through complete single neuron reconstruction.

From previous studies, GABAergic neurons in the DR and MR are thought to innervate other neuron types in the same raphe nucleus to modulate function through disynaptic pathways. Suppression of DR GABAergic neurons could alleviate the acquisition of social avoidance by promoting the activity of serotonergic neurons (*Challis et al., 2013*). MR GABAergic neurons modulate hippocampal theta rhythm by innervating MR serotonergic neurons, and indirectly regulate hippocampal ripple activity by inhibiting MR non-GABAergic neurons (*Li et al., 2005*; *Wang et al., 2015*). However, DR GABAergic neurons also regulate thermogenesis through long-range projections to the DMH, BST and related areas (*Schneeberger et al., 2019*). Given the observed vast range of projections from GABAergic neurons in the DR and MR, there appears to be an underappreciated potential functional role of GABAergic projection neurons in the DR and MR. Whether the same GABAergic neurons in the raphe nuclei participate in the direct and indirect pathways simultaneously needs further investigation.

There are several caveats in our viral tracing techniques and data analysis. The monosynaptic rabies virus tracing technique might only label a fraction of inputs; moreover, the labeling might be biased toward specific neuron types and affected by many factors (*Callaway and Luo, 2015*). For output analysis, the axonal terminals and fibers of passage are not distinguished. Whether all labeled neurons project to all target brain regions or part of them are not known. It might cause results to be somewhat different from the true projection strength and projection pattern. Furthermore, given the variability of viral transduction in individual samples, the data were normalized as a proportion to interpret the tracing results, but these quantification results might underestimate the connections of small brain regions.

In summary, we constructed a comprehensive whole-brain atlas of inputs and outputs of glutamatergic and GABAergic neurons in the DR and MR, which revealed similar input patterns but divergent projection patterns. The differences in connectivity patterns are related to specific regulatory processes of specific functions. Since the whole-brain connections of genetically targeted neurons are key factors in characterizing cell types, our results could inform the generation of whole-brain cell atlases that are under ongoing effort. Our work contributes to the foundation for exploring the relationships among cell heterogeneity, anatomical connectivity, and behavior function of the raphe nuclei. This connectivity atlas has focused on the neural circuits of specific neuron types in the DR and MR, and there is a long way to systematically construct brain's wiring diagram of more precise resolution.

## Materials and methods

**Key resources table**

| Reagent type (species) or resource | Designation | Source or reference | Identifiers | Additional information |
|---|---|---|---|---|
| Antibody | Anti-Vglut3 (Rabbit polyclonal) | Thermo Fisher Scientific | RRID: AB_2736782 | (1:200) |
| Antibody | Anti-rabbit Alexa Fluor 594 (Donkey polyclonal) | Invitrogen | RRID:AB_141637 | (1:500) |

*Continued on next page*

*Continued*

| Reagent type (species) or resource | Designation | Source or reference | Identifiers | Additional information |
|---|---|---|---|---|
| Genetic reagent (*Mus musculus*) | *Vglut2-Cre* | The Jackson Laboratory | Cat# JAX:016963, RRID:IMSR_JAX:016963 | *Slc17a6tm2(cre)Lowl/J* |
| Genetic reagent (*Mus musculus*) | *Gad2-Cre* | The Jackson Laboratory | Cat# JAX:010802, RRID:IMSR_JAX:010802 | *Gad2tm2(cre)Zjh/J* |
| Recombinant DNA reagent | rAAV2/9-EF1α-DIO-His-TVA-BFP | BrainVTA Co., Ltd. | http://brainvta.tech/html/AAV_services/ | $2 \times 10^{12}$ viral genomes/ml |
| Recombinant DNA reagent | rAAV2/9- EF1α-DIO-RG | BrainVTA Co., Ltd. | http://brainvta.tech/html/AAV_services/ | $2 \times 10^{12}$ viral genomes/ml |
| Recombinant DNA reagent | RV-ΔG-EnvA-GFP | BrainVTA Co., Ltd. | http://brainvta.tech/plus/list.php?tid=114 | $2 \times 10^{8}$ infectious units/ml |
| Recombinant DNA reagent | AAV-DIO-EYFP | BrainVTA Co., Ltd. | http://brainvta.tech/html/AAV_services/ | $2 \times 10^{12}$ viral genomes/ml |
| Software, algorithm | Amira | FEI, Mérignac Cedex, France | RRID:SCR_007353 | |
| Software, algorithm | Imaris | Bitplane, Zurich, Switzerland | RRID:SCR_007370 | |
| Software, algorithm | Fiji | https://imagej.net/Fiji | RRID:SCR_002285 | |
| Software, algorithm | Python 3.6.4 | http://www.python.org | RRID:SCR_008394 | |
| Software, algorithm | R 4.0.3 | R-project | RRID:SCR_001905 | |
| Software, algorithm | Matlab | Mathworks, MA | RRID:SCR_001622 | |
| Software, algorithm | Graphpad Prism | GraphPad, CA | RRID:SCR_002798 | |

## Animals

In this study, adult *Vglut2-Cre* (*Slc17a6tm2(cre)Lowl/J*, stock number: 016963) and *Gad2-Cre* (*Gad2tm2(cre)Zjh/J*, stock number: 010802) mice purchased from the Jackson Laboratory were used. All mice were housed in an experiment environment with 12 hr light/dark cycle, 22°C ± 1 °C temperature, 55% ± 5% humidity and food and water ad libitum. All animal experiments were approved by the Institutional Animal Care and Use Committee at HUST-Suzhou Institute For Brainsmatics and were conducted in accordance with relevant guidelines.

## Stereotaxic injections

For retrograde monosynaptic tracing, 150 nl adeno-associated helper virus (AAV helper) was injected into the DR (bregma: –4.6 mm, lateral: 0 mm, ventral: –3.0 mm) or MR (bregma: –4.6 mm, lateral: 0 mm, ventral: –4.25 mm) in *Vglut2-Cre* and *Gad2-Cre* mice. Three weeks later, 200 nl RV-ΔG-EnvA-GFP ($2 \times 10^{8}$ infectious units/ml) was injected into the same site. One week later, the mice were used for sample preparation. The AAV helper was a 1:2 mixture of rAAV2/9-EF1α-DIO-His-TVA-BFP ($2 \times 10^{12}$ viral genomes/ml) and rAAV2/9- EF1α-DIO-RG ($2 \times 10^{12}$ viral genomes/ml). For antegrade tracing, 50 nl AAV-DIO-EYFP ($2 \times 10^{12}$ viral genomes/ml) was injected into the DR (bregma: –4.6 mm, lateral: 0 mm, ventral: –3.0 mm) or MR (bregma: –4.6 mm, lateral: 0 mm, ventral: –4.25 mm) in adult *Vglut2-Cre* and *Gad2-Cre* mice. Three weeks later, the mice were used for sample preparation. All viral tools were produced by BrainVTA Co., Ltd.

## Histology

The anesthetized mice were intracardially perfused with 0.01 M PBS (Sigma-Aldrich Inc), followed by 4 % paraformaldehyde (Sigma-Aldrich Inc) in 0.01 M PBS. The brains were excised and post-fixed in 4 % paraformaldehyde at 4 °C for 24 hr.

For whole-brain imaging, the brains were embedded following a previously described workflow (*Ren et al., 2018b*). Briefly, each brain was rinsed overnight at 4 °C in 0.01 M PBS and dehydrated in a graded ethanol series (50, 70, and 95% ethanol, changing from one concentration to the next every 1 hr at 4 °C). After dehydration, the brains were immersed in a graded glycol methacrylate (GMA)

series (Ted Pella Inc), including 0.2 % SBB (70, 85, and 100 % GMA for 2 hr each and 100 % GMA overnight at 4 °C). Finally, the samples were impregnated in a prepolymerization GMA solution for 3 days at 4 °C and embedded in a vacuum oven at 35 °C for 24 hr.

For immunohistochemistry, the mouse brains were sectioned into 70-μm-thick coronal slices using the vibrating slicer (VT1200S, Leica). The selected coronal sections were blocked in 0.01 M PBS containing 5 % bovine serum albumin and 0.3 % Triton-X 100 for 1 hr, and then incubated with the primary antibodies (12 hr at 4 °C): anti-Vglut3 (1:200, Rabbit, Thermo Fisher Scientific, PA 5–77432). After rinsing, the sections were incubated with the fluorophore-conjugated secondary antibody (1:500, Donkey-Anti-Rabbit, Invitrogen, Alexa Fluor 594) for 2 hr at 37 °C. The antibodies were diluted in the same block solution.

For in situ hybridization, the anesthetized mice were intracardially perfused with 1× PBS DEPC and 4 % paraformaldehyde (including 1‰ DEPC) (Boster Biological Technology Co., Ltd.). The brains were excised and post-fixed in 4 % paraformaldehyde (including 1‰ DEPC) for 24 hr. Then, the samples were immersed in 30 % sucrose (1× PBS DEPC) for 24 hr, then embedded by Tissue-Tek O.C.T. Compound (Sakura). The 1× PBS DEPC was prepared by 1× PBS (PH:7.2–7.4) and DEPC at the ratio of 1:1,000. The embedded samples were sectioned into 12 μm coronal slices using the freezing microtome (CM1950, Leica). The target region with high hybridization efficiency and specificity was selected from the RNA sequence of the tissues to design hybrid probe. The probes were designed by spatial FISH., Ltd. A reaction chamber was prepared on the coronal slices and fixed with 4 % paraformaldehyde, then dehydrated and denatured with methanol. And the hybrid buffer was added to the reaction chamber for overnight incubation at 37 °C. Next, the ligase reaction system was added to the reaction chamber. And the Phi29DNA polymerase amplification system was used for rolling circle amplification and signal amplification. Finally, fluorescence hybridization probe was added to the reaction chamber for in situ hybridization.

## Imaging and image preprocessing

For whole-brain high-resolution imaging, the virus-labeled and GMA resin-embedded samples were imaged with propidium iodide (PI) simultaneously staining cytoarchitecture landmarks using our home-made fMOST system at a resolution of $0.32 \times 0.32 \times 2 \ \mu m^3$. The acquired two-channel raw data were processed through mosaic stitching and illumination correction to piece together into entire coronal sections as previously described (*Gong et al., 2016*). Each channel dataset of single brain sample contains approximately 5500 coronal slices. For in situ hybridization and immunohistochemistry staining, the coronal sections were imaged using the confocal microscope (TCS SP8, Leica). For starter cells, the samples were sectioned into 70-μm-thick coronal slices using the vibrating slicer (VT1200S, Leica) and imaged using the automated slide scanner (VS120 Virtual Slide, Olympus).

## Data processing

### Registration

To quantify and integrate the whole-brain connections, the coordinates of the soma of input neurons and high-resolution image stack of labeled outputs were registered to Allen CCFv3 using the transformation parameters acquired by the previously described methods (*Ni et al., 2020*). Briefly, we segmented several brain regions as landmarks through cytoarchitecture references, such as the outline, caudoputamen, medial habenula, lateral ventricle, and third ventricle. Based on these landmarks, we performed affine transformation and symmetric image normalization in Advanced Normalization Tools (ANTS) to acquire transformation parameters.

### Nomenclature of the brain regions

Demarcation and annotation of brain regions were based on Allen CCFv3. The superior central nucleus raphe (CS) is equivalent to the MR with the consultation of the mouse brain atlas by Paxinos and Franklin (*Paxinos and Franklin, 2012*). Based on Allen CCFv3's hierarchy of brain regions, since there were few or no input neurons and projections in numerous brain regions, we collapsed some brain regions to their 'parent' region as appropriate. Therefore, we divided the whole-brain into 117 brain regions (see *Supplementary file 1*) and identified 71 brain regions for analysis (areas that have small proportion of connections are merged into 'Others'). The STR-NA, PAL-NA, TH-NA, HY-NA, MB-NA,

P-NA, and MY-NA refer to the non-annotated area in the striatum, pallidum, thalamus, hypothalamus, midbrain, pons, and medulla, respectively.

## Detection and quantification of whole-brain inputs
We automatically identified and localized the soma of input neurons using NeuroGPS (*Quan et al., 2013*) and manually checked the results to eliminate indiscernible mistakes. Next, we warped the soma coordinates to Allen CCFv3 using the transformation parameters from the aforementioned registration. Finally, we calculated the number and proportion of input neurons in each brain region of interest to generate the quantified whole-brain inputs.

## Detection and quantification of whole-brain outputs
We generated quantified whole-brain outputs by taking following steps:

We resampled the image stack of labeled neural structures to isotropic 1 μm, segmented the outline of brain, set the intensity of pixels outside the outline to 0, and used the transformation parameters of the aforementioned registration to warp them to Allen CCFv3 at 1 μm scaling. Then, we manually segmented injection sites on registered coronal sections.

To detect projection signal from background, each registered coronal section was background subtracted, Gaussian filtered, and threshold segmented to binary image. The background image $I$ was calculated as $I = min(I, background)$ followed by ten convolutions with the averaging template of $9 \times 9$ size, where $I$ is the gray level of coronal section and the *background* is an proximate estimated background intensity (*Quan et al., 2013*). The size of gaussian filter was $5 \times 5$. The filtered image was binarized by $max(4\sqrt{I}, threshold)$ , where *threshold* was the value calculated by the Yen method that clipped to the predetermined threshold range (*Yen et al., 1995*).

The whole-brain images were divided into $10 \times 10 \times 10$ μm³ grids. In each division, we calculated signal density by the definition of the sum of detected pixels divided by the sum of all pixels in a three-dimensional grid, therefore generated a three-dimensional signal density matrix of 10 μm voxel resolution. Subsequently, we calculated the computational path based on the signal density matrix using multistencils fast marching algorithm, and removed the voxels that could not back-track to the injection site or back-track to injection site with low confidence (*Oh et al., 2014*; *Hassouna and Farag, 2007*; *Liu et al., 2018*). The confidence of the path was defined as the proportion of back-tracking points of the path located in the foreground voxel, with the foreground voxel referring to the voxel whose signal density was greater than a threshold. Finally, we manually inspected the results and removed the remaining confusing noise voxels.

The outputs were quantified as projection signal volume in each brain region normalized by signal volume across whole brain (with exclusion of the injection site). Since the soma and dendrites of labeled neurons contributed numerous signals in the injection site, we excluded the injection site for more accuracy.

## Visualization and statistical analysis
The Amira software (v6.1.1, FEI) and Imaris software (v9.5.0, Bitplane) were used to visualize the inputs and outputs. To compare the inputs to glutamatergic and GABAergic neurons in the DR and MR across brain regions, we performed one-way analysis of variance (ANOVA) followed by multiple comparisons with Tukey's test. To compare the outputs of glutamatergic and GABAergic neurons in the same nucleus across brain regions, we performed one-way ANOVA. To explore the similarities and variances of inputs/outputs of brain regions connected with glutamatergic and GABAergic neurons in the DR and MR, we performed correlation analysis and hierarchical cluster analysis. These processes were performed using MATLAB (v2017a, MathWorks) and Python 3.6.4. To compare the whole-brain inputs across samples, we performed hierarchical clustering and bootstrapping using pvclust that is a package of R (*Suzuki and Shimodaira, 2006*). All histograms were generated using GraphPad Prism (v.6.0, GraphPad).

## Acknowledgements

We thank H Ni, M Ren, X Wang for help with experiments, data analysis, and constructive comments, and the other members of MOST group of Britton Chance Center for Biomedical Photonics and HUST-Suzhou Institute for Brainsmatics for help with experiments and data acquisition. We thank G

Cao from Huazhong Agricultural University for help with validation of specific labeling. This work was financially supported by the National Natural Science Foundation of China (Grant Nos. 91749209, 61890953, 81827901), the Science Fund for Creative Research Group of China (Grant No.61721092), and the CAMS Innovation Fund for Medical Sciences (CIFMS 2019-I2M-5-014).

## Additional information

### Funding

| Funder | Grant reference number | Author |
| --- | --- | --- |
| National Natural Science Foundation of China | 91749209 | Qingming Luo |
| National Natural Science Foundation of China | 61890953 | Hui Gong |
| National Natural Science Foundation of China | 81827901 | Anan Li |
| National Natural Science Foundation of China | 61721092 | Qingming Luo |
| Chinese Academy of Medical Sciences | CIFMS 2019-I2M-5-014 | Qingming Luo |

The funders had no role in study design, data collection and interpretation, or the decision to submit the work for publication.

### Author contributions

Zhengchao Xu, Data curation, Formal analysis, Methodology, Software, Visualization, Writing – original draft, Writing – review and editing; Zhao Feng, Data curation, Software; Mengting Zhao, Software, Visualization; Qingtao Sun, Formal analysis, Validation; Lei Deng, Tao Jiang, Investigation; Xueyan Jia, Data curation, Visualization; Pan Luo, Data curation, Validation; Wu Chen, Formal analysis; Ayizuohere Tudi, Validation; Jing Yuan, Xiangning Li, Investigation, Resources; Hui Gong, Conceptualization, Resources, Writing – review and editing; Qingming Luo, Conceptualization, Funding acquisition; Anan Li, Methodology, Project administration, Writing – original draft, Writing – review and editing

### Author ORCIDs

Zhengchao Xu http://orcid.org/0000-0002-9462-1638
Zhao Feng http://orcid.org/0000-0001-5035-7655
Mengting Zhao http://orcid.org/0000-0003-2037-9129
Qingtao Sun http://orcid.org/0000-0003-2167-7750
Lei Deng http://orcid.org/0000-0003-2774-5611
Xueyan Jia http://orcid.org/0000-0002-1221-6357
Tao Jiang http://orcid.org/0000-0002-4487-299X
Pan Luo http://orcid.org/0000-0002-3923-2111
Wu Chen http://orcid.org/0000-0001-5673-4888
Ayizuohere Tudi http://orcid.org/0000-0002-2451-2903
Jing Yuan http://orcid.org/0000-0001-9050-4496
Xiangning Li http://orcid.org/0000-0002-3747-2824
Hui Gong http://orcid.org/0000-0001-5519-6248
Qingming Luo http://orcid.org/0000-0002-6725-9311
Anan Li http://orcid.org/0000-0002-5877-4813

### Ethics

All animal experiments were approved by the Institutional Animal Care and Use Committee at HUST-Suzhou Institute For Brainsmatics (S20190601) and were conducted in accordance with relevant guidelines.

### Decision letter and Author response

Decision letter https://doi.org/10.7554/eLife.65502.sa1

Author response https://doi.org/10.7554/eLife.65502.sa2

## Additional files

### Supplementary files

• Supplementary file 1. Nomenclature and abbreviations of brain regions.

• Supplementary file 2. Quantification and comparison of whole-brain inputs to glutamatergic and GABAergic neurons in the DR and MR.

• Supplementary file 3. Quantification and comparison of whole-brain outputs of glutamatergic and GABAergic neurons in the DR and MR.

• Transparent reporting form

### Data availability

The analysis results and data have been uploaded in Supplementary Files 1,2 and 3. To present and share the TB-sized raw data, we developed an interactive website (http://atlas.brainsmatics.org/a/xu2011).

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
