## [Editor Report]

Using viral labeling method in combination with the fMOST imaging technology, the authors constructed a whole brain connectivity atlas of two subclasses, glutamatergic and GABAergic, of neurons in the dorsal raphe and median raphe nuclei. This study will be of interest to many neuroscientists who study neural circuits and cell type-specific functions.

---

## [Decision Letter]

**Decision letter after peer review:**

Thank you for submitting your article "Whole-brain connectivity atlas of glutamatergic and GABAergic neurons in mouse dorsal and median raphe nucleus" for consideration by *eLife*. Your article has been reviewed by 3 peer reviewers, including Hongwei Dong as the Reviewing Editor and Reviewer #1, and the evaluation has been overseen by Lu Chen as the Senior Editor.

Essential revisions:

All three reviewers find your work of potential interest with a recommendation for publication if the manuscript is revised appropriately. However, there is a cluster of revisions and clarifications warranted before publication. Several common concerns raised by all reviewers are listed here:

1. VGluT2-Cre line needs to be carefully clarified for its characterization and expression in the MR and DR. Utility of VgluT2-cre needs to be justified, since the field has generally considered the Vglut3 as the major DR glutamatergic population, and there are numerous previous papers using Vglut3-cre instead of Vglut2-cre lines. With this consideration, the reviewers agree that adding the Vglut3 dataset would substantially improve the manuscript and make it much more relevant to the field. Nevertheless, we understand it is a substantial amount of work to be added and it will take a longer time to finish the revision. Please carefully consider your revision plan.

2. Injection sites need to be carefully characterized and analyzed. Please see reviewer 3's comments "The quantification throughout the manuscript refers to the relative proportion of inputs or outputs for each cell population and nucleus. The manuscript would be strengthened by also including total cell counts for starter cells in each group, as well as total numbers of input neurons. For example, is the Vglut2 population in DR much larger than the Gad2 population, and do DR Vglut2 neurons receive more inputs in total than DR Gad2 neurons?"

3. The manuscript's English needs to be proofread extensively for readability and clarity.

4. Please address all other concerns from three reviewers as listed below.

*Reviewer #1 (Recommendations for the authors):*

The manuscript's English needs to be proofread extensively. The Discussion needs to be better organized.

*Reviewer #2 (Recommendations for the authors):*

1. While GAD2-Cre would generally label all mature GABAergic neurons, VGluT2-Cre only labels a major population of glutamatergic neurons. Others are VGluT1/3 positive and could have distinct connections and functions, as briefly mentioned in the discussion. In title and content where glutamatergic was more broadly indicated with current studies, it needs to be clarified upfront.

2. For both input and output analysis, the accuracy is still limited by current available approaches. For input analysis, we do not yet know if RV-mediate approach would cause any biased toward specific populations. For output analysis, there are both false-positive (en passant projection vs bona fide targeting) and false-negative (are the entire projections detected for all labeled neurons?) possibilities that needs to be discussed.

*Reviewer #3 (Recommendations for the authors):*

Please provide number of mice used for the input tracing, it is not mentioned in the text, methods section, nor the figure legend.

Has this exact helper and RV virus been used in previous studies? It is known that the TVA expression can be leaky and result in non cell-type specific expression. Due to this it is advisable to do control experiments in wild-type mice to exclude non-specific expression of starter cells.

Last sentence of the abstract should be changed, it is too strong to claim that the study provides insights into behavioral functions. I agree that anatomical connectivity may suggest the possibility of certain functional roles, but the wording should reflect that uncertainty.

In Figure 5A, it's valuable to show the differences in cell distribution topography within LH. The left-most panel would be improved if you could incorporate the antero-posterior information in a 3D illustration, since now it is quite difficult to really see the differences in the separate input-populations. Furthermore, the right-most panel somewhat undermines your conclusion of "MR GABAergic neurons were assembled more laterally than the input neurons to MR glutamatergic neurons on the whole" as the graph shows very subtle differences. Perhaps adding a line to denote the median midline distance for each population would emphasize the differences you'd like to show.

I appreciated the online resources showing coronal sections of rabies tracing. However, it was very slow to actually load the single coronal images and I was not able to load them in high resolution. Perhaps lowering the file sized would enable users to more efficiently explore this resource.

Overall, the text could be improved in terms of grammar and verb tense inconsistently used throughout.

[Editors' note: further revisions were suggested prior to acceptance, as described below.]

Thank you for submitting your article "Whole-brain connectivity atlas of glutamatergic and GABAergic neurons in the mouse dorsal and median raphe nuclei" for consideration by *eLife*. Your article has been reviewed by 3 peer reviewers, one of whom is a member of our Board of Reviewing Editors, and the evaluation has been overseen by Lu Chen as the Senior Editor. The reviewers have opted to remain anonymous.

Essential revisions:

1) All reviewers appreciated that the authors have addressed their major concerns. Reviewer 3 has further recommendations with a hope to further improve the clarity of the paper. Please revise the manuscript accordingly.

*Reviewer #1 (Recommendations for the authors):*

The authors have addressed critiques of all reviewers. I have no more concerns.

*Reviewer #2 (Recommendations for the authors):*

The revised manuscript has sufficiently addressed my major concerns. I don't have further comment for it.

*Reviewer #3 (Recommendations for the authors):*

I appreciate the author's addition of more refined characterisation of mouse lines used, as well as the virus injection sites. This puts the starting point of their input and output tracing into context and is crucial to include. It was also good to see the proportion of starter cells specific to the DR and MR, as the broad expression of these cell types was a concern for targeting specificity.

1) Line 56 "Given that numerous Vglut3+ neurons in the DR and MR are also serotonergic (Huang et al., 2019; Pinto et al., 2019; Sos et al., 2017), the present study focused on the connectivity of VGluT2^+^ neurons in the DR and MR."

Since the authors are using the rationale of Vglut3/Serotonin co-expression being a reason to only target VGluT2^+^ neurons in their study, please provide evidence that the Vglut2 population does not co-express serotonin (references or own quantification).

2) Line 122-124 the authors write, "We performed in situ hybridization to characterize the specificity of labeled starter cells in the Vglut2-Cre mice and found that they were Vglut2 positive, with a few simultaneously being Vglut3 positive."

I would like to see an actual quantification of this VGluT2^+^/Vglut3+ subpopulation, how many cells did you count and % double positive.

3) Line 449, "Our work could form the foundation for exploring the relationships among cell heterogeneity, anatomical connectivity, and behavior function of the raphe nuclei."

Please rewrite as "Our work contributes to the foundation for …" since this study is not the first to explore cell types and connectivity of these nuclei.

4) Please rewrite the very last sentence in the discussion, it is very clunky.

---

## [Author Response]

Essential revisions:All three reviewers find your work of potential interest with a recommendation for publication if the manuscript is revised appropriately. However, there is a cluster of revisions and clarifications warranted before publication. Several common concerns raised by all reviewers are listed here:1. VGluT2-Cre line needs to be carefully clarified for its characterization and expression in the MR and DR. Utility of VgluT2-cre needs to be justified, since the field has generally considered the Vglut3 as the major DR glutamatergic population, and there are numerous previous papers using Vglut3-cre instead of Vglut2-cre lines. With this consideration, the reviewers agree that adding the Vglut3 dataset would substantially improve the manuscript and make it much more relevant to the field. Nevertheless, we understand it is a substantial amount of work to be added and it will take a longer time to finish the revision. Please carefully consider your revision plan.

As suggested, we verified the *Vglut2-Cre* line for its characterization and expression in the DR and MR. We crossed the Cre driver line mice with reporter line mice respectively (*Vglut2-Cre: LSL-H2B-GFP* mice and *Gad2-Cre: LSL-H2B-GFP* mice) to get the distribution of endogenous gene expression of VGluT2^+^ and Gad2+ neurons (Figure 1—figure supplement 1). As a result, there are populations of VGluT2^+^ and Gad2+ neurons in the DR and MR.

We performed in situ hybridization to characterize the specificity of labeled starter cells in the *Vglut2-Cre* mice and found that they were Vglut2 positive, with a few simultaneously being Vglut3 positive (Figure 1B,C; Figure1—figure supplement 3), which was confirmed by immunohistochemical staining (Figure 1—figure supplement 4).

Moreover, we tried a lot to acquire the Vglut3 dataset. Since adding the Vglut3 dataset required quantities of *Vglut3-Cre* mice and substantial time to perform labeling, imaging, and analysis, we did not get enough datasets for analysis at present. Here, we chose to clarify that we focused on the connections of VGluT2^+^ neurons in the DR and MR (Line 41-43).

2. Injection sites need to be carefully characterized and analyzed. Please see reviewer 3's comments "The quantification throughout the manuscript refers to the relative proportion of inputs or outputs for each cell population and nucleus. The manuscript would be strengthened by also including total cell counts for starter cells in each group, as well as total numbers of input neurons. For example, is the Vglut2 population in DR much larger than the Gad2 population, and do DR Vglut2 neurons receive more inputs in total than DR Gad2 neurons?"

As for the inputs, we added the number of input neurons in Supplementary File 2. And samples from the same batch of virus tracing experiments were treated as validation datasets for analysis of starter cells. We counted the starter cells and input neurons, then calculated the on-target rate of starter cells and the ratio of input neurons to starter cells (Figure 1—figure supplement 2C). Compared with experiment datasets, they have consistent input patterns (Figure 1—figure supplement 2D). The monosynaptic rabies tracing technique could be biased toward specific neuron types and affected by many factors. The ratio of input neurons to starter cells variate in a vast range (Callaway and Luo. 2015. The Journal of Neuroscience, 35: 8979–8985). The larger population of specific neuron types might not indicate that they receive more inputs.

As for the outputs, we manually segmented the injection region and calculated the proportion of signal in the injection region within DR/MR (Figure 4—figure supplement 1). And we counted the labeled neurons in the DR and MR and calculated the percentage (Supplementary File 3). And we added the volume of the projection signal in Supplementary File 3.

3. The manuscript's English needs to be proofread extensively for readability and clarity.

We invited two native English experts to proofread the manuscript's English and revise the whole manuscript.

4. Please address all other concerns from three reviewers as listed below.

All the concerns were addressed point by point.

Reviewer #2 (Recommendations for the authors):1. While GAD2-Cre would generally label all mature GABAergic neurons, VGluT2-Cre only labels a major population of glutamatergic neurons. Others are VGluT1/3 positive and could have distinct connections and functions, as briefly mentioned in the discussion. In title and content where glutamatergic was more broadly indicated with current studies, it needs to be clarified upfront.

We clarified that we focused on the connectivity of Vglut2+ neurons in the DR and MR (Line 41-43).

2. For both input and output analysis, the accuracy is still limited by current available approaches. For input analysis, we do not yet know if RV-mediate approach would cause any biased toward specific populations. For output analysis, there are both false-positive (en passant projection vs bona fide targeting) and false-negative (are the entire projections detected for all labeled neurons?) possibilities that needs to be discussed.

We revised the discussion section and pointed out the limitations in line 413-421.

“There are several caveats in our viral tracing techniques and data analysis. The monosynaptic rabies tracing technique might only label a fraction of inputs; moreover, the labelling might be biased toward specific neuron types and affected by many factors (Callaway and Luo, 2015). For output analysis, the axonal terminals and fibers of passage are not distinguished. Whether all labeled neurons project to all target brain regions or part of them are not known. It might cause results to be somewhat different from the true projection strength and projection pattern. Furthermore, given the variability of viral transduction in individual samples, the data were normalized as a proportion to interpret the tracing results, but these quantification results might underestimate the connections of small brain regions.”

Reviewer #3 (Recommendations for the authors):Please provide number of mice used for the input tracing, it is not mentioned in the text, methods section, nor the figure legend.

As for the quantification of inputs, the data were replicated in 4 mice. We provided the number of mice used for the input tracing in the figure legend of Figure 2.

Has this exact helper and RV virus been used in previous studies? It is known that the TVA expression can be leaky and result in non cell-type specific expression. Due to this it is advisable to do control experiments in wild-type mice to exclude non-specific expression of starter cells.

The exact AAV helper and RV virus have been used in previous studies (Sun et al. Nature Neuroscience, 2019; Zhao et al. Scientific Reports, 2020, 10:12209). To evaluate leakage expression of the virus, we performed control experiments in wild-type mice. As a result, there were few neurons infected by the AAV helper virus (BFP) and the RV (GFP) only at the injection site, but there were no cells expressing GFP in known upstream brain regions. These data showed that there was no transsynaptic transmission to input neurons (Figure 1—figure supplement 5). Furthermore, as mentioned in Line 130-132, when we quantified the whole brain inputs, the neurons in the injection region were excluded.

Last sentence of the abstract should be changed, it is too strong to claim that the study provides insights into behavioral functions. I agree that anatomical connectivity may suggest the possibility of certain functional roles, but the wording should reflect that uncertainty.

We have revised this sentence as, “This connectivity atlas further elucidates the anatomical architecture of the raphe nuclei, which could facilitate better understanding of their behavioral functions.”.

In Figure 5A, it's valuable to show the differences in cell distribution topography within LH. The left-most panel would be improved if you could incorporate the antero-posterior information in a 3D illustration, since now it is quite difficult to really see the differences in the separate input-populations. Furthermore, the right-most panel somewhat undermines your conclusion of "MR GABAergic neurons were assembled more laterally than the input neurons to MR glutamatergic neurons on the whole" as the graph shows very subtle differences. Perhaps adding a line to denote the median midline distance for each population would emphasize the differences you'd like to show.

We have revised Figure 5A and incorporated the antero-posterior information in a 3D illustration. And for the right-most panel of Figure 5A, we added a line to denote the median midline distance for each population as suggested, and we moved it to Figure 5—figure supplement 1C. As the differences in the graph are not especially obvious, but the distribution of MR GABAergic neurons was more lateral on the whole, we revised the description, “Regarding the LH, the input neurons to MR GABAergic neurons seemed to be distributed more laterally than the input neurons to MR glutamatergic neurons on the whole”.

I appreciated the online resources showing coronal sections of rabies tracing. However, it was very slow to actually load the single coronal images and I was not able to load them in high resolution. Perhaps lowering the file sized would enable users to more efficiently explore this resource.

To better present our raw data, we resampled the raw data and uploaded the data with different resolutions (0.32×0.32×50 µm^3^ and 3.2×3.2×50 µm^3^) to the website (http://atlas.brainsmatics.org/a/xu2011).

Overall, the text could be improved in terms of grammar and verb tense inconsistently used throughout.

We invited two native English experts to proofread the manuscript's English and revise the whole manuscript.

[Editors' note: further revisions were suggested prior to acceptance, as described below.]

Reviewer #3 (Recommendations for the authors):I appreciate the author's addition of more refined characterisation of mouse lines used, as well as the virus injection sites. This puts the starting point of their input and output tracing into context and is crucial to include. It was also good to see the proportion of starter cells specific to the DR and MR, as the broad expression of these cell types was a concern for targeting specificity.1) Line 56 "Given that numerous Vglut3+ neurons in the DR and MR are also serotonergic (Huang et al., 2019; Pinto et al., 2019; Sos et al., 2017), the present study focused on the connectivity of VGluT2^+^ neurons in the DR and MR."Since the authors are using the rationale of Vglut3/Serotonin co-expression being a reason to only target VGluT2^+^ neurons in their study, please provide evidence that the Vglut2 population does not co-express serotonin (references or own quantification).

In the previous studies, single-cell RNA sequencing was performed to dissect molecular neuron subtypes in the DR. The Slc17a6(Vglut2)-expressing glutamatergic subtype lacked expression of Slc6a4 and Tph2, which indicated that the VGluT2^+^ neurons in the DR were distinct from serotonergic neurons in the DR (Huang et al., 2019). The previous studies also revealed that the VGluT2^+^ neurons in the MR were distinct from serotonergic neurons in the MR via immunohistochemistry (Szőnyi et al., 2019). These results indicate that the Vglut2 population does not co-express serotonin. Thus, we revised the sentence as:

Line 41-44, “Given that numerous Vglut3+ neurons in the DR and MR are also serotonergic (Huang et al., 2019; Pinto et al., 2019; Sos et al., 2017), while VGluT2^+^ neurons in the DR and MR were distinct from the serotonergic neurons (Huang et al., 2019; Szőnyi et al., 2019), the present study focused on the connectivity of VGluT2^+^ neurons in the DR and MR.”

2) Line 122-124 the authors write, "We performed in situ hybridization to characterize the specificity of labeled starter cells in the Vglut2-Cre mice and found that they were Vglut2 positive, with a few simultaneously being Vglut3 positive."I would like to see an actual quantification of this VGluT2^+^/Vglut3+ subpopulation, how many cells did you count and % double positive.

In response to previous comments, we tried various experiments to characterize and validate the specificity of labeled starter cells in the Vglut2-Cre mice. Further analysis of these results might be able to answer the new questions. We performed in situ hybridization to characterize the specificity of labeled starter cells in the Vglut2-Cre mice. In a 12 μm thick coronal slice, we counted 21 stater cells in the DR and 19.05% (4/21) were double positive; we counted 12 cells in the MR in a coronal slice and 8.33% (1/12) were double positive. Furthermore, we performed immunohistochemical staining to validate the specificity of starter cells, we counted 32 starter cells in the DR and 15.63% (5/32) were also Vglut3 positive; we counted 38 starter cells in the MR and 10.53% (4/38) were also Vglut3 positive.

As shown in Figure 1 B, Figure 1—figure supplement 3, and Figure 1—figure supplement 4, these results indicated that there were a few Vglut2 positive neurons were also Vglut3 positive both in the DR and MR.

3) Line 449, "Our work could form the foundation for exploring the relationships among cell heterogeneity, anatomical connectivity, and behavior function of the raphe nuclei."Please rewrite as "Our work contributes to the foundation for …" since this study is not the first to explore cell types and connectivity of these nuclei.

As suggested, we have rewritten the sentence as:

Line 428-430, "Our work contributes to the foundation for exploring the relationships among cell heterogeneity, anatomical connectivity, and behavior function of the raphe nuclei."

4) Please rewrite the very last sentence in the discussion, it is very clunky.

We have rewritten the sentence as:

Line 430-432, “Ths connectivity atlas has focused on the neural circuits of specific neuron types in the DR and MR, and there is a long way to systematically construct brain’s wiring diagram of more precise resolution.”